Resource

# Transcriptome analyses in infertile men reveal germ cell–specific expression and splicing patterns

Lara M Siebert-Kuss[1],*, Henrike Krenz[2],*, Tobias Tekath[2], Marius Wöste[2], Sara Di Persio[1], Nicole Terwort[1], Margot J Wyrwoll[3], Jann-Frederik Cremers[4], Joachim Wistuba[1], Martin Dugas[2,5], Sabine Kliesch[4], Stefan Schlatt[1], Frank Tüttelmann[3], Jörg Gromoll[1], Nina Neuhaus[1],†, Sandra Laurentino[1],†

The process of spermatogenesis—when germ cells differentiate into sperm—is tightly regulated, and misregulation in gene expression is likely to be involved in the physiopathology of male infertility. The testis is one of the most transcriptionally rich tissues; nevertheless, the specific gene expression changes occurring during spermatogenesis are not fully understood. To better understand gene expression during spermatogenesis, we generated germ cell–specific whole transcriptome profiles by systematically comparing testicular transcriptomes from tissues in which spermatogenesis is arrested at successive steps of germ cell differentiation. In these comparisons, we found thousands of differentially expressed genes between successive germ cell types of infertility patients. We demonstrate our analyses' potential to identify novel highly germ cell–specific markers (TSPY4 and LUZP4 for spermatogonia; HMGB4 for round spermatids) and identified putatively misregulated genes in male infertility (*RWDD2A, CCDC183, CNNM1, SERF1B*). Apart from these, we found thousands of genes showing germ cell–specific isoforms (including *SOX15, SPATA4, SYCP3, MKI67*). Our approach and dataset can help elucidate genetic and transcriptional causes for male infertility.

## Introduction

Spermatogenesis is a complex process by which spermatogonia undergo differentiation, becoming spermatocytes, which, after undergoing meiosis, originate haploid spermatids and finally sperm. Disturbances in spermatogenesis, which cause male infertility, can range from arrest at different steps during germ cell differentiation to the complete absence of germ cells, known as a Sertoli cell–only (SCO) phenotype.

To understand the gene expression profiles of specific testicular cell types and, thus, to gain information about changes in gene expression during spermatogenesis that may lead to male infertility, previous studies have taken advantage of samples with distinct histological phenotypes of male infertility. Specifically, prior studies used samples matched by cellular composition and also performed comparative microarray analyses of samples differing in the presence of one specific germ cell type (von Kopylow et al, 2010; Chalmel et al, 2012; Lecluze et al, 2018). For example, in a study that compared testicular tissues with SCO and spermatogonial arrest phenotypes, which only differ in the presence of spermatogonia, von Kopylow et al (2010) were able to identify transcripts specifically expressed by spermatogonia. They identified the spermatogonial markers *FGFR3* and *UTF1*, which are currently considered specific markers for different spermatogonial subpopulations (Guo et al, 2018; Sohni et al, 2019; Di Persio et al, 2021). Chalmel et al (2012) expanded on this approach by including samples from (pre)pubertal and adult arrest phenotypes, thereby extracting the transcriptional profiles of additional germ cell types. These studies demonstrated that comparing distinct arrest phenotypes allows for identifying transcripts expressed at specific stages of germ cell differentiation during normal spermatogenesis (von Kopylow et al, 2010; Chalmel et al, 2012).

Currently, technological developments such as RNA sequencing (RNA-seq) enable an unbiased and more comprehensive analysis of the transcriptome. Specifically, single-cell RNA sequencing (scRNA-seq) of human testicular tissues has revolutionized germ cell–specific RNA profiling by allowing the identification of cell type–specific gene expression patterns (Guo et al, 2018; Hermann et al, 2018; Wang et al, 2018; Sohni et al, 2019; Di Persio et al, 2021). However, scRNA-seq offers sparser data compared with conventional bulk RNA-seq and, by sequencing only the near-poly-A extremities of the transcripts, generates limited information on transcriptional isoforms (Tekath & Dugas, 2021). Therefore, RNA-seq

[1]Centre of Reproductive Medicine and Andrology, Institute of Reproductive and Regenerative Biology, University of Münster, Münster, Germany    [2]Institute of Medical Informatics, University of Münster, Münster, Germany    [3]Institute of Reproductive Genetics, University of Münster, Münster, Germany    [4]Department of Clinical and Surgical Andrology, Centre of Reproductive Medicine and Andrology, University Hospital of Münster, Münster, Germany    [5]Institute of Medical Informatics, Heidelberg University Hospital, Heidelberg, Germany

Correspondence: Sandra.Laurentino@ukmuenster.de
*Lara M Siebert-Kuss and Henrike Krenz are joint first authors
†Nina Neuhaus and Sandra Laurentino are joint senior authors

provides the most complete capture of the transcriptome, including all transcripts obtained through post-transcriptional processing.

Notably, the testis presents unusually high levels of these post-transcriptional events, including alternative splicing (AS) (Kan et al, 2005). AS enables the production of different transcripts and potentially different proteins from a single gene. Splice-site variants in some genes (the follicle-stimulating and luteinizing hormone receptor genes) have been linked to human male infertility (Song et al, 2002; Bruysters et al, 2008). However, it remains to be elucidated which role different transcript isoforms play in regulating spermatogenesis and how different isoforms are involved in the pathology of male infertility. Knowledge of the changes in isoforms that result from AS during human spermatogenesis would open a new avenue for identifying so far unknown causes of male infertility.

The role of different genes and their variants in testicular physiopathology is far from being elucidated. In this study, we aimed at generating whole transcriptome profiles of human testicular germ cells. For the first time, we combined total RNA-seq of distinct pathological phenotypes with published scRNA-seq data to unveil the transcriptome profiles of male germ cells and determined changes in AS during human spermatogenesis. Using this setup, we evaluated the functional consequences of a pathogenic variant in a male infertility case, demonstrating the potential of the outlined approach.

# Results

## Clinical and histological evaluation of the patient cohort

To study germ cell–specific whole transcriptome changes during human spermatogenesis, we carefully selected histologically characterized testicular biopsies (Tables 1 and S1) presenting homogenous phenotypes of azoospermia (azoospermia = absence of sperm in the ejaculate, n = 16), namely, no germ cells present in the testicular tissue (SCO, n = 3);

arrests at the spermatogonial (SPG, n = 4), spermatocyte (SPC, n = 3), or round spermatid (SPD, n = 3) levels; and complete spermatogenesis as controls (CTR, n = 3) (Fig 1A and B). Except in the CTR samples with complete spermatogenesis, no sperm was found via microscopic examination of the mechanically dissociated biopsies (Table 1).

## Genetic characterization of the patient cohort

No patients showed chromosomal abnormalities except for one (spermatid arrest patient SPD-3) who had a low-grade XXY mosaicism (47,XXY[2]/46,XY[28]). A control patient (CTR-1) was previously diagnosed during routine genetic diagnostics with the heterozygous *CFTR* variants c.1521_1523delCTT p.(Phe508del) and c.2991G>C p.(Leu997Phe), suggesting compound heterozygosity, which represents the cause for a congenital absence of *vas deferens* (CBAVD) in this man. By analyzing whole exome sequencing (WES) data of our patients, we identified a heterozygous missense variant in *SYCP3* (patient SCO-2), which is predicted to potentially affect splicing (NM_153694.4: c.551A>C p.(Lys184Thr)). We identified a heterozygous missense variant in *PLK4* with a CADD score of 28.8 (NM_014264.5 c.950C>T p.(Pro317Leu)) and the heterozygous splice-site variant NM_021951.3 c.355-4C>T p.? in *DMRT1*, which might also have an impact on splicing (patient SPD-1). A patient with spermatocyte arrest (SPC-1) was identified in a parallel study to carry a homozygous deletion affecting the complete *SYCE1* gene (Wyrwoll et al, 2022). A patient with spermatogonial arrest (SPG-1) was in parallel identified with the heterozygous synonymous variant NM_004959.5 c.990G>A p.(Glu330=) in *NR5A1*, which affects the last base of exon 5 and is also predicted to alter splicing (Wyrwoll et al, 2022).

## Transcriptome analyses recapitulate the phenotypic and genetic characteristics of the patient cohort

We sequenced total RNA obtained from testicular biopsies, including all transcript isoforms deriving from alternative splicing.

**Table 1. Clinical characteristics of the patient groups.**

| Patient groups | Karyotype | Histological parameters of tubules | | | | | | | Hormonal parameters (normal range) | | | Sperm mTESE |
|---|---|---|---|---|---|---|---|---|---|---|---|---|
| | | Score | % ES | % RS | % SC | % SG | % SCO | % TS | FSH (1–7 U/l) | LH (2–10 U/l) | T(>12 nmol/l) | |
| SCO (n = 3) | 46,XY | 0 | 0 | 0 | 0 | 0 | 98.7 (±1.5) | 1.3 (±1.5) | 13.3 (±4.2) | 5.8 (±2.6) | 13.7 (±3.4) | No |
| SPG (n = 4) | SPG-1, SPG-2, SPG-3: 46,XY, SPG-4: n.d. | 0 | 0 | 0 | 0 | 31.0 (±34.6) | 34.3 (±20.7) | 35.0 (±20.6) | 20.4 (±14.2) | 13.4 (±9.7) | 16.2 (±6.9) | No |
| SPC (n = 3) | 46,XY | 0 | 0 | 0 | 89.3 (±11.0) | 4.7 (±4.6) | 1.0 (±1.0) | 5.3 (±5.5) | 5.7 (±1.3) | 5.7 (±4.5) | 9.9 (±2.4) | No |
| SPD (n = 3) | SPD-1, SPD-2: 46,XY, SPD-3a | 0 | 0 | 28.3 (±2.3) | 59.3 (±18.0) | 3.0 (±2.0) | 1.7 (±2.9) | 8.7 (±14.2) | 7.4 (±0.9) | 3.7 (±0.5) | 18.7 (±5.7) | No |
| CTR (n = 3) | 46,XY | 8–10 | 87.3 (±8.6) | 3.3 (±2.5) | 8.7 (±5.7) | 0 | 0 | 1.0 (±1.0) | 2.5 (±1.3) | 2.6 (±1.0) | 24.7 (±2.2) | Yesb |

Data are presented as mean ± SD. Percentage of tubules with the most advanced germ cell type present: elongated spermatids (%ES), round spermatids (%RS), spermatocytes (%SPC), spermatogonia (%SPG), Sertoli cell–only phenotype (%SCO), or tubular shadows (%TS). Score refers to Bergmann and Kliesch score (Bergmann & Kliesch, 2010). Hormonal parameters for follicle-stimulating hormone (FSH), luteinizing hormone (LH) and testosterone (T).
aPatient SPD-3 had a low number of XXY karyotype mosaicism (47,XXY[2]/46,XY[28]).
bTesticular sperm extraction (TESE) results: CTR-1 had 100/100 sperm, CTR-2 had an average of 89/100 sperm; no TESE result available for CTR-3 because the reason for surgery was a suspected malignant tumor. SCO, Sertoli cell–only; SPG, spermatogonial arrest; SPC, spermatocyte arrest; SPD, round spermatid arrest; CTR, control spermatogenesis; n.d., not determined.

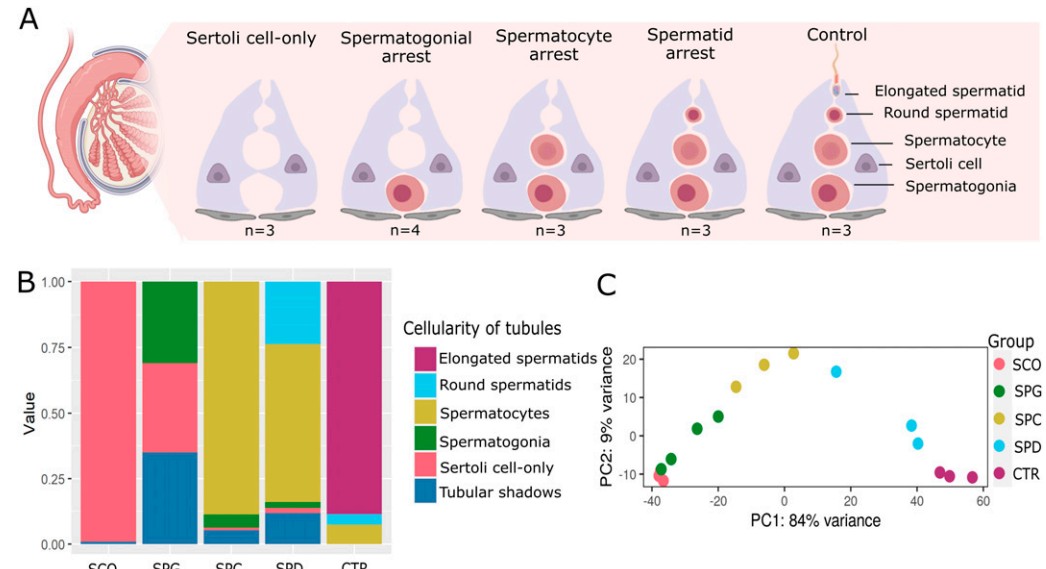

**Figure 1. Cellular composition of the human testicular biopsies.**
**(A)** Schematic illustration depicts the cellular composition of the testicular biopsies with Sertoli cell–only phenotype, arrest at the spermatogonial (SPG), spermatocyte (SPC), and spermatid (SPD) stage and samples with complete spermatogenesis, which were used as controls (CTR). **(B)** Stacked bar plots represent the proportional cellularity of round seminiferous tubules ranked according to the most advanced germ cell type in the tubule. The cellularity of samples from each group is averaged. **(C)** Principal component analysis (PCA) plot depicts clustering of the total RNA–sequenced samples based on the top 500 genes.

After RNA-seq, principal component analysis (PCA) organized the spermatogenic arrest samples in a consecutive order (Fig 1C), mirroring their sequential spermatogenic phenotypes.

To evaluate the extent to which the identified exome variants influence the testicular transcriptome, we analyzed the identified variants in the total RNA-seq data of the respective patients. In line with the homozygous deletion of *SYCE1*, we detected no RNA of *SYCE1* in SPC-1 in comparison to SPC-2 and SPC-3. We found that the heterozygous synonymous variant in *NR5A1* of patient SPG-1 led to an alternative 5′ splice site in the affected exon 5 (Fig 2). This originates from a transcript with an in-frame deletion of 48 nucleotides. For all other variants, which were predicted to affect splicing, no alternative splice sites were identified.

### Comparative analysis reveals germ cell–specific transcriptome profiles

We aimed at generating germ cell–specific expression profiles to study transcriptome changes throughout spermatogenesis. To this end, we systematically performed differential gene expression (DEG) analysis between groups of different cellularities, representing the four main differentiation steps of male germ cells: SCO versus SPG, SPG versus SPC, SPC versus SPD, and SPD versus CTR (Fig 3A). This revealed between 839 and 4,138 DEGs in the four group comparisons (FDR < 0.05 and absolute log$_2$ FC ≥ 1).

In the SCO versus SPG comparison, most transcript changes were due to the increased expression of 2,073 genes in SPG samples (Table S2). These DEGs also remained highly expressed in other groups containing spermatogonia (SPC, SPD, CTR), indicating that most of these transcripts originate from the presence of spermatogonia (Fig 3B). Indeed, among the highly expressed genes were

well-known spermatogonial genes such as *MAGEA4* and *FGFR3* (Table S3). The most prominent changes in gene expression were found when comparing SPG with SPC samples (Table S4). The 2,886 genes that were high in expression included spermatocyte-specific genes like *AURKA* and *OVOL1* (Table S3). The same genes also showed high expression in SPD and CTR samples and low to absent expression in SPG and SCO. This indicates that these genes are specific to spermatocytes rather than the result of gene expression alterations in other cell types. When comparing SPC with SPD samples, we found 2,345 highly expressed genes in SPD samples (Table S5), including spermiogenesis marker genes *TNP1* and *PRM1* (Table S3). These genes also showed higher expression in CTR samples and lower expression in samples lacking spermatids (SPC, SPG, SCO), in accordance with their spermatid-specific expression pattern. The most subtle changes in gene expression were detected when comparing SPD with control samples (Table S6), in which the presence of elongated spermatids is the only histological difference. Genes with increased expression in CTR samples (776) showed lower expression levels in the spermatogenic arrest samples (SPD, SPC, SPG, SCO) and, among others, included genes associated with the sperm flagellum like *CATSPER3* and *TEKT2* (Table S3).

### Novel germ cell–specific marker genes and their expression at single-cell resolution

To identify novel germ cell–specific marker genes, we focused on the top 120 DEGs, ranked by their log$_2$ FC, with elevated expression in SPG, SPC, SPD, and CTR samples (Fig 3C–F). We evaluated all top DEGs per group comparison for their germ cell specificity in our published scRNA-seq dataset of three patients with complete spermatogenesis (Di Persio et al, 2021) (Fig 3G) and in one additional

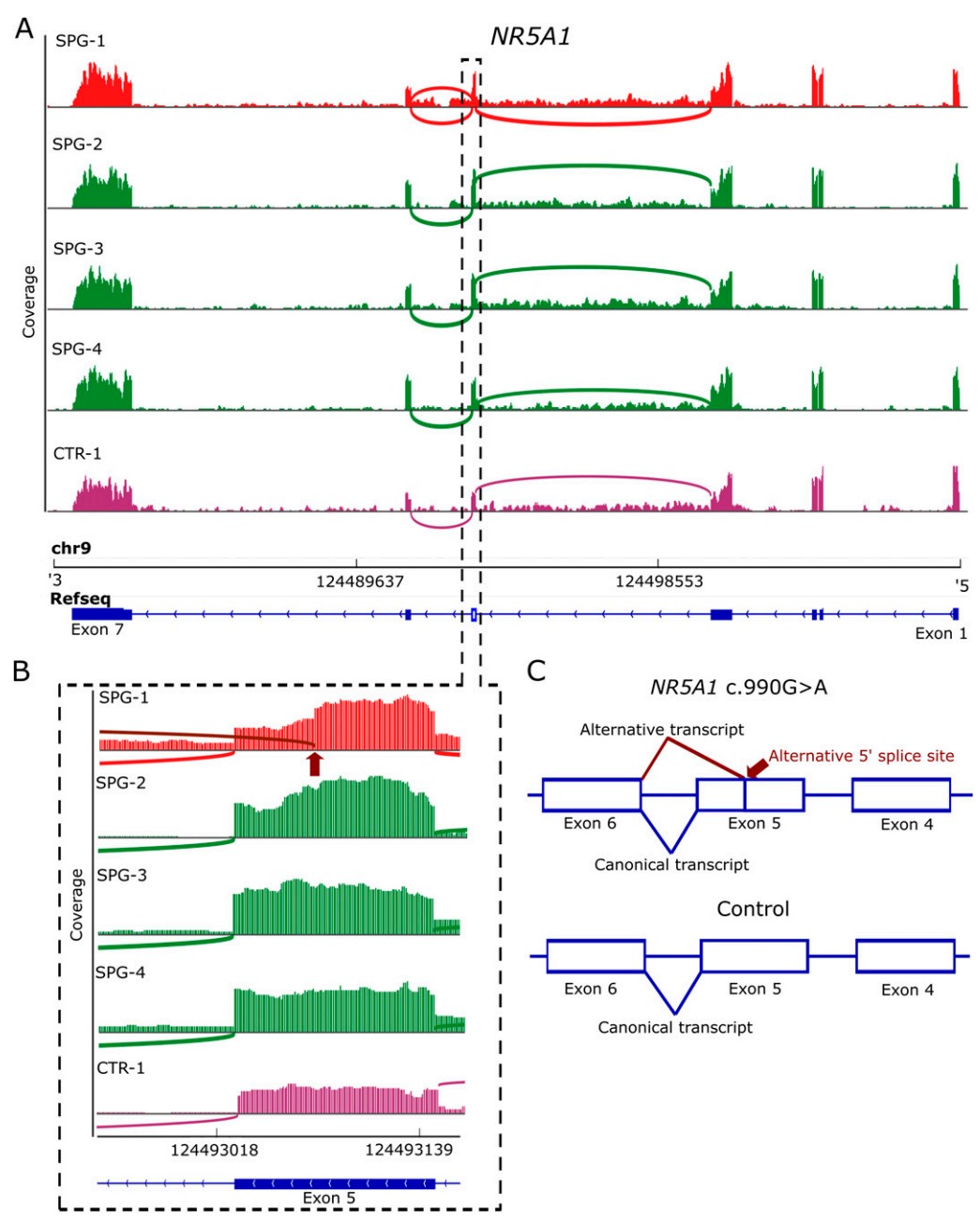

**Figure 2. Alternative 5′ splice site in exon 5 of *NR5A1* in one patient with spermatogonial arrest (SPG-1).**
**(A)** Sashimi plots depicting the read coverage as bars across the genomic location of *NR5A1* in patient SPG-1 carrying the heterozygous synonymous variant NM_004959.5 c.990G>A p.(Glu330=) (red) in comparison to the other SPG patients (green) and one control patient (CTR-1, purple). Arcs represent the splice junctions of exon 5 according to the sequencing reads. Boxes indicate the coding region and larger boxes the untranslated regions in the Refseq. **(B)** Zoom into the coverage plots for exon 5 shows the alternative 5′ splice site in SPG-1 (dark red arc and arrow), which is not present in the other patients and which leads to a decrease in coverage in the last 48 nucleotides of exon 5. **(C)** Schematic illustration of the splicing consequence in the coding region because of the heterozygous synonymous variant in comparison to the other patients without the pathogenic variant serving as controls. In the patient carrying the variant, both the canonical transcript and a transcript with a 48 nucleotide deletion are present.

scRNA-seq dataset also of three patients with complete spermatogenesis (Hermann et al, 2018). We found that 12–27% of the top DEGs per group comparison were absent or showed very low expression levels in the scRNA-seq datasets evaluated (Table S7). Among the undetected genes were long non-coding and read-through RNAs of two neighboring genes. An average of 85 ± 9% of

genes were represented in the scRNA-seq datasets and displayed highly germ cell–specific expression patterns (Fig S1).

Among the genes with highly germ cell–specific expression (Fig S2), we identified potential new marker genes for spermatogonia (Fig 3H; *leucine zipper protein 4* (*LUZP4*); *testis-specific protein Y–linked 4* (*TSPY4*); *anomalous homeobox* (*ANHX*)), spermatocytes

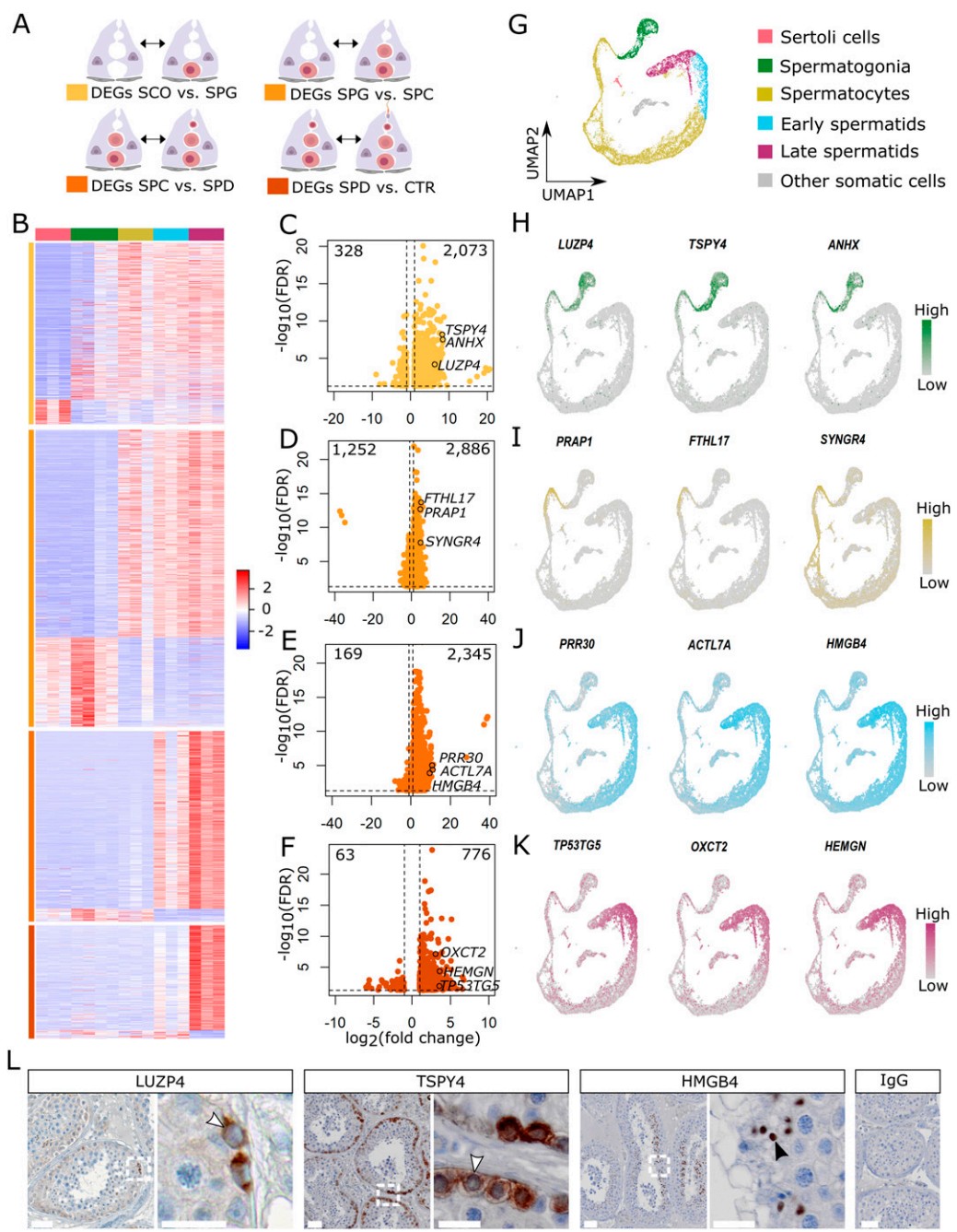

**Figure 3. Examination of germ cell–specific gene expression.**
**(A)** Schematic illustration of the group comparisons and the respective color codes of their differentially expressed genes (DEGs). **(B)** The heat map displays the normalized expression counts of the DEGs (rows) of each group comparisons across all samples (columns) scaled via a row Z-score. Red = increased; blue = decreased. **(C, D, E, F)** Volcano plots of the increased and decreased genes in samples with (C) spermatogonial, (D) spermatocyte, (E) and spermatid arrest and in (F) complete spermatogenesis. **(G)** UMAP plot depicts 15,546 cells integrated from three patients with obstructive azoospermia and complete spermatogenesis (Di Persio et al, 2021). **(H, I, J, K)** Feature plots show the expression of three novel genes for (H) spermatogonia, (I) spermatocytes, (J) round spermatids, and (K) elongated spermatids at single-cell level. **(L)** Micrographs showing immunohistochemical stainings for LUZP4, TSPY4, and HMGB4 in testicular tissue with full spermatogenesis (n = 3). Arrow heads in the inlays indicate positive spermatogonia (white) and round spermatids (black). IgG control shows no staining. Scale bars = 50 $\mu$m for micrographs and 20 $\mu$m for inlays. Data information: genes with a false discovery rate (FDR) < 0.05 and a log$_2$ fold change (FC) ≥ 1 were considered DEGs based on Wald test and adjusted with Benjamini–Hochberg.

(Fig 3I; *proline rich acidic protein 1* (*PRAP1*); *ferritin heavy chain like 17* (*FTHL17*); *synaptogyrin 4* (*SYNGR4*)), round spermatids (Fig 3J; *proline rich 30* (*PRR30*); *actin like 7A* (*ACTL7A*); *high mobility group box 4* (*HMGB4*)), and elongated spermatids (Fig 3K; *TP53 target 5*

(*TP53TG5*); *3-oxoacid CoA-transferase 2* (*OXCT2*); *hemogen* (*HEMGN*)). We evaluated the expression at the protein level for three of the identified marker genes (TSPY4, LUZP4, HMGB4) and found that these markers are indeed expressed specifically in the

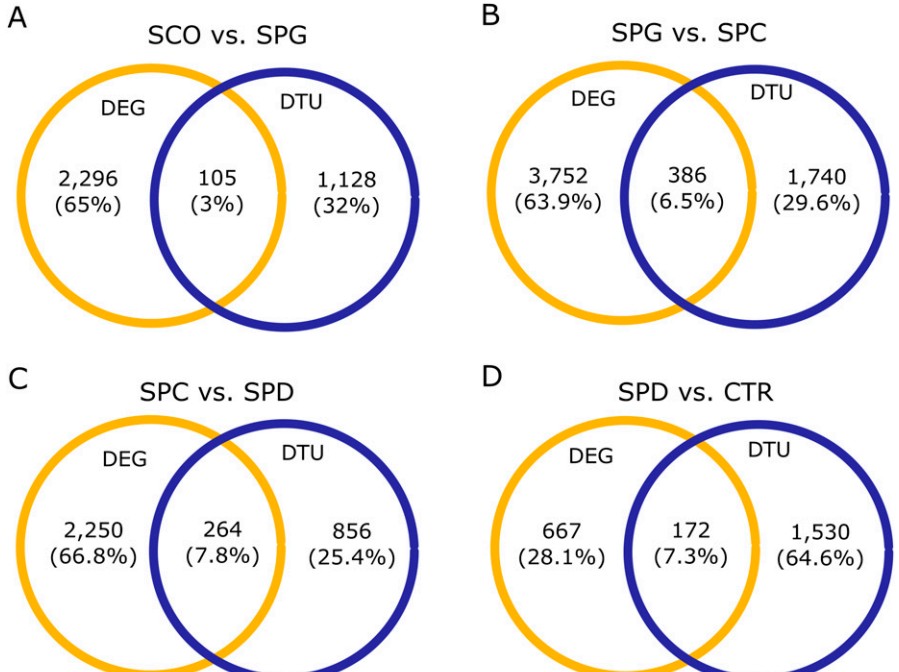

**Figure 4. Comparison of differentially expressed gene (DEG) and differential trascript usage (DTU) gene numbers in all four group comparisons.**
**(A, B, C, D)** Venn diagrams display number and proportion of genes that are differentially expressed, have a DTU event, or both in the (A) Sertoli cell–only versus SPG, (B) SPG versus SPC, (C) SPC versus SPD, and (D) SPD versus CTR group comparisons. Yellow = differential gene expressions, blue = DTU genes.

expected germ cell types in control samples with full spermatogenesis (Fig 3L). We further characterized the spermatogonial specificity of our newly identified spermatogonial marker TSPY4. Co-localization with the pan-spermatogonial marker MAGEA4 revealed that TSPY4 is expressed in 88 ± 5.2% of MAGEA4+ cells (Fig S3A). To evaluate whether TSPY4 is a marker for undifferentiated spermatogonia, we co-immunolocalized TSPY4 with the pan-undifferentiated spermatogonial marker UTF1. We found that an average of 85 ± 5.6% of undifferentiated spermatogonia also express TSPY4 (Fig S3B).

### Alternative splicing is uncoupled from gene expression

To study alternative splicing, we performed a differential transcript usage (DTU) analysis between all four group comparisons. DTU analysis calculates and compares the proportional contributions (referred to as "usage") of transcripts to the overall expression of a gene. A gene has a DTU event, that is, is a DTU gene, when at least two of its transcripts are differentially used between two groups. We found between 1,062 and 2,153 DTU genes in each of the four comparisons (Tables S8–S11). By comparing DTU genes to DEGs, we found an overlap of less than 8% in all four comparisons, indicating that the expression of most genes is regulated either at the pre- or the post-transcriptional level (Fig 4) and that only few genes are regulated at both levels. Furthermore, we found that the proportion of DEGs to DTU genes in all group comparisons was 2:1 (Fig. 4A–C), except for SPD versus CTR, where this ratio was inversed with more DTU genes than DEGs (Fig 4D).

### DEGs and DTU genes are involved in different biological pathways

We used Ingenuity Pathway Analysis (IPA) to evaluate the molecular functions of the DEGs and DTU genes in the different germ cell

types. In line with the small overlap between the DEG and DTU gene sets, we found minor overlaps between the top 20 significantly enriched molecular functions of DEGs and DTU genes in all four groups (Fig 5). Both gene sets contained genes involved in organization of cytoskeleton/cytoplasm, microtubule dynamics, apoptosis, necrosis, and segregation of chromosomes. IPA analysis on DEGs highlighted functional enrichment annotations that can be attributed to the most advanced germ cell type in each group comparison (e.g., development of stem cells, segregation of chromosomes) (Fig 5A). In comparison to the functional annotations of DEGs, 26% of molecular functions of the DTU genes overlapped across the four group comparisons (Fig 5B). Among the overlapping terms were microtubule dynamics, organization of cytoplasm, and cytoskeleton. More general biological functions (e.g., RNA metabolism, cell survival) were enriched among the DTU genes in each group comparison. To further classify the biological pathways enriched among DEGs and DTU genes, we performed pathway analysis via the Reactome Knowledgebase (Gillespie et al, 2022), which confirmed that germ cell–specific and general pathways are enriched among DEGs (Fig.S4) and DTU genes (Fig.S5), respectively.

### Germ cell type–dependent splicing is an additional layer of gene regulation in the germline

To study alternatively spliced transcripts, we investigated the transcript biotypes of selected DTU genes. In comparison to the proportional distribution of transcript biotypes annotated in GENCODE (Frankish et al, 2019), we found that most of the DTU events, regardless of the group comparison, result in protein-coding transcripts (Fig.6A). In the comparison between SPG and SPC

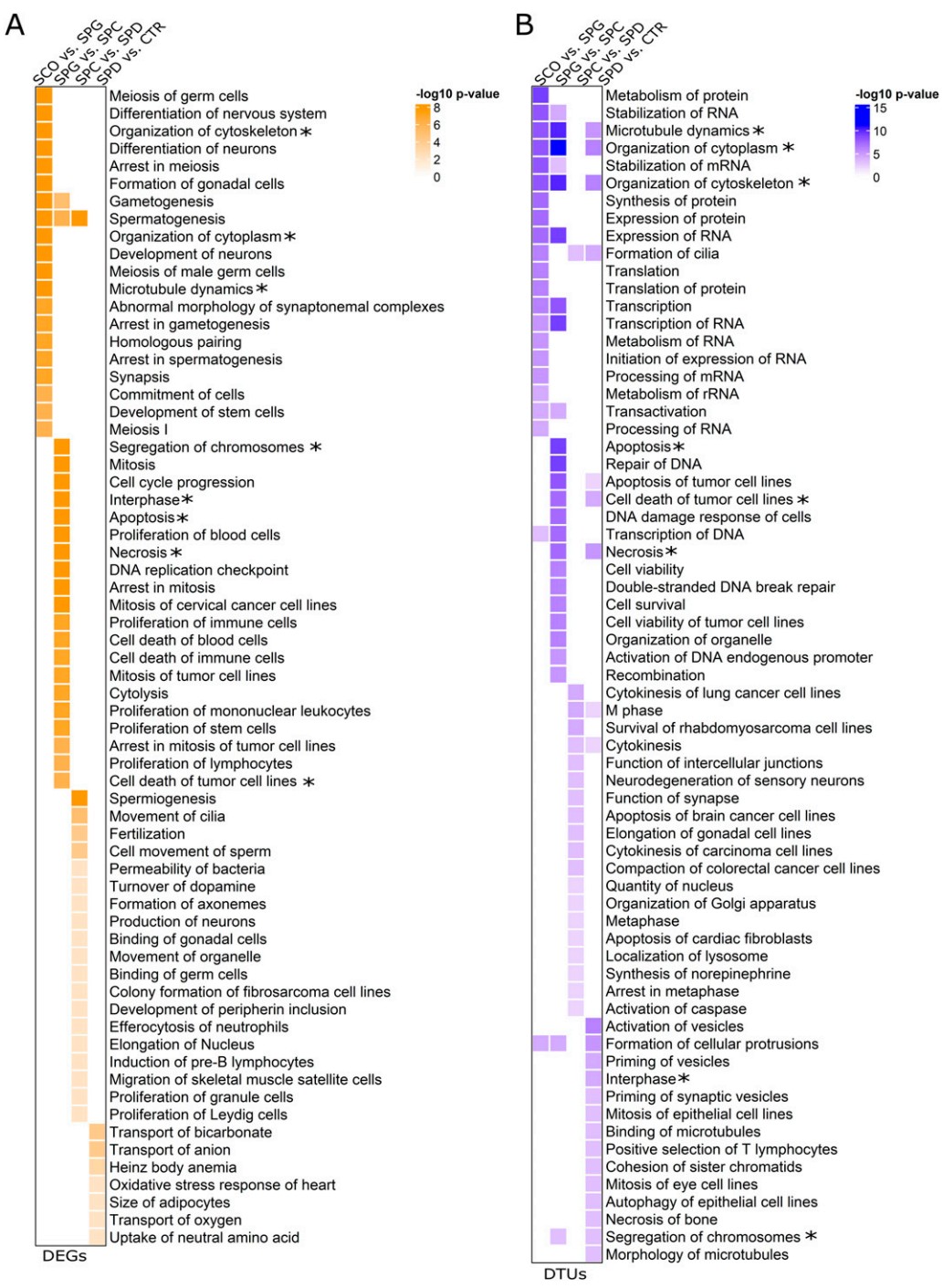

**Figure 5. Molecular functions of differentially expressed genes (DEGs) and differential trascript usage (DTU) genes.**
**(A, B)** Heat maps displaying the molecular functions revealed by IPA of all (A) DEGs and (B) DTU genes per group comparisons according to the $-\log_{10}$ P-values. The top 20 molecular functions of each group comparison with P-values < 0.01 were included. * Molecular functions enriched in both the DEG and DTU gene sets.

samples (Fig 6B), two protein-coding isoforms of *SRY-Box Transcription Factor 15* (*SOX15*) displayed differential usage without changes in gene expression (Fig 6C). Although *SOX15-201* (ENST00000250055.3) was the predominant isoform, with an average usage of 48% in SPC samples, SPG samples predominantly used the *SOX15-202* isoform (ENST00000538513.6), which has an

alternative 5′ splice site in the 5′ UTR region. Reverse transcriptase quantitative PCR (RT-qPCR) analysis of *SOX15* replicated both the differential usage of *SOX15-201* and the equal gene expression levels between SPG and SPC samples (Fig S6A–D). *Spermatogenesis associated 4* (*SPATA4*) also showed a switch in usage for its protein-coding isoforms *SPATA4-201* (ENST00000280191.7) and *SPATA4-203*

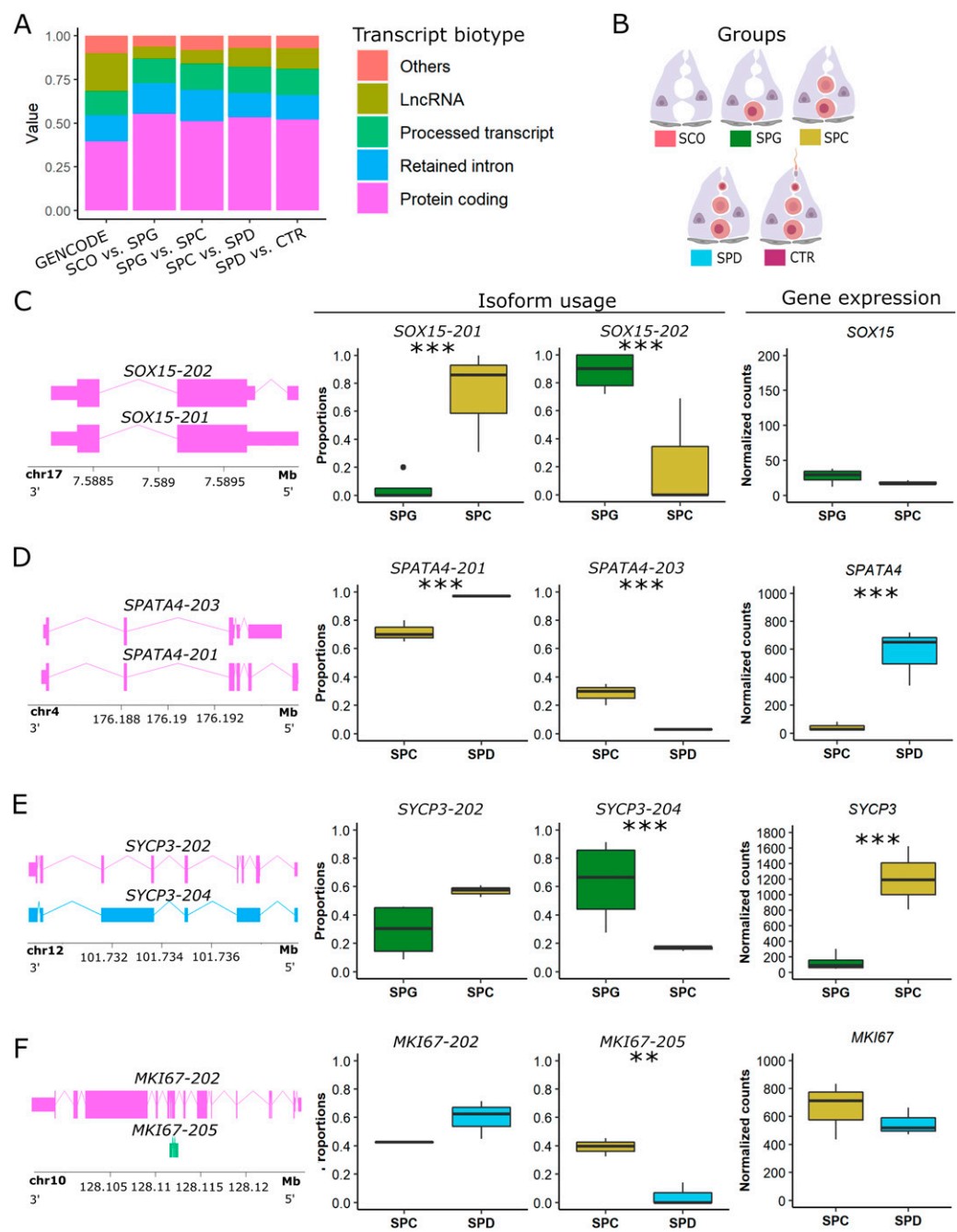

**Figure 6. Transcript biotypes with differential transcript usage (DTU) events.**
**(A)** Stacked bar plots represent the relative amount of different transcript biotypes with DTU events in each of the four group comparisons compared with the transcript biotype annotation from the GENCODE release 36 genome annotation based on the GRCh38.p13 genome reference (Frankish et al, 2019). **(B)** Schematic illustration of the groups and the respective color codes. **(C, D, E, F)** Schematic representation of the transcript isoforms with a DTU event, which predominantly contribute to the relative change in isoform usage (box plots of proportion), independent of gene expression (boxplots of normalized counts) in (C) *SOX15*, (D) *SPATA4*, (E) *SYCP3*, and (F) *MKI67*. P-values refer to specific transcripts that significantly drive the change in isoform usage in genes with an overall significant change in transcript usage. In (C, D, E, F), data are represented as median (center line), upper/lower quartiles (box limits), 1.5× interquartile range (whiskers), and outliers (points). Likelihood ratio test: **$P \leq 0.01$, ***$P \leq 0.001$. Exons/coding region = boxes, UTR = smaller boxes, introns = lines. SPG: n = 4; SPC: n = 3, SPD: n = 3.

(ENST00000515234.1) in the comparison of SPC versus SPD samples (Fig 6D). SPC samples showed a significantly decreased usage of *SPATA4-201* and a significantly increased usage of *SPATA4-203*, whereas SPD samples exclusively used the *SPATA4-201* isoform. These two isoforms use alternative stop sites. In contrast to *SOX15*,

*SPATA4* was also a DEG in this group comparison and had a higher expression level in SPD samples.

Intriguingly, the second largest group of biotypes with DTU events were retained introns (Fig 6A). For *synaptonemal complex protein 3* (*SYCP3*), we found a significantly increased usage of the

retained intron isoform *SYCP3-204* (ENST00000478139.1) in SPG samples, whereas SPC samples had an increased usage of the protein-coding isoform *SYCP3-202* (ENST00000392924.2; Fig 6E). In this group comparison, *SYCP3* showed increased expression in SPC samples. We confirmed the increased usage of the retained intron isoform together with the decreased expression in SPG samples by RT-qPCR analysis (Fig S6E–H). A switch in usage from coding to non-coding transcripts was also observed for *marker of proliferation Ki-67* (*MKI67*), which did not show changes in gene expression (Fig 6F). However, the protein-coding isoform *MKI67-202* (ENST00000368654.8) was less expressed in SPC samples in comparison to SPD samples. In contrast, the processed transcript isoform *MKI67-205* (ENST00000484853.1) showed significantly increased usage in SPC samples and decreased usage in SPD samples.

### Identification of putative infertility genes

By making use of samples derived from infertility patients, we aimed at identifying genes related to male infertility that have so far been understudied. We analyzed genes with enriched expression in SCO samples compared with SPG, SPG to SPC, SPC to SPD, and SPD to CTR (genes in blue in Tables S2 and S4–S6). Analysis via the Reactome Knowledgebase revealed that the most significant biological pathways enriched among the up-regulated genes in the SCO group were GABA-related processes, for example, MECP2 regulates the transcription of genes involved in GABA signaling and GABA synthesis (Fig S7A). A significant enrichment of genes involved in the immune response was found up-regulated in SPG samples, involving pathways for interferon and cytokine signaling (Fig S7B). The most significant pathway enriched in up-regulated genes of SPC samples was the regulation of IGF transport and uptake by IGF-binding proteins (Fig S7C). In contrast, up-regulated genes in the SPD group were most significantly enriched for metabolic pathways, including rRNA processing in the mitochondrion and electron transport from NADPH to ferredoxin (Fig S7D). We then evaluated the cell type–specific expression of the most severe 50 putatively misregulated genes in our scRNA-seq dataset. In all group comparisons, genes showed predominant expression in the somatic cells (Fig S7E–H). Some genes stood out, such as those that were up-regulated in SCO (*RWDD2A*, *CCDC183*, *CNNM1*) or SPD (*SERF1B*) samples but, according to scRNA-seq, displayed a germ cell–specific or meiotic-specific expression pattern, respectively (Fig S7E and H). According to normal tissue data available in the Genotype-Tissue Expression (GTEx) portal (release v8, accessed on July 2022), the exons of *RWDD2A*, *CCDC183*, *CNNM1*, and *SERF1B* are predominantly expressed in the testis (Fig S8).

## Discussion

Reports on gene expression patterns in the testis are accumulating rapidly, but a complete picture of the transcriptome of human germ cells has remained unexplored. Here, we demonstrate that the progression of human male germ cell differentiation is accompanied by major transcript dynamics, including germ cell type–dependent transcription and splicing events; these splicing events result in germ cell type–dependent transcript isoforms.

Because of the use of microarrays in previous studies, the full spectrum of transcriptome profiles, including isoform information, has remained largely unknown. Our systematic analysis of total RNA from testicular biopsies with well-defined, distinct germ cell compositions allowed us to identify highly germ cell–specific genes that, to our knowledge, have not been previously associated with the respective germ cell types in humans (Table S12).

*In silico* analyses of these putative infertility-related genes pointed to potentially misregulated pathways. To date, none of the identified germ cell–specific genes that were significantly up-regulated in our infertility groups (*RWDD2A*, *CCDC183*, *CNNM1*, *SERF1B*) had been linked to male reproductive health, rendering genes revealed in this study potential candidates to investigate their role in male infertility. Future studies will be necessary to conclude whether the different expression of the genes is due to misregulation or is secondary to the absence of specific spermatogenic cells.

The transcriptional output of a gene depends not only on the level of RNA expression but also on post-transcriptional processing of RNA transcripts, for instance, through AS, which allows a single gene to originate different transcripts and potentially different proteins (Baralle & Giudice, 2017). Although it is well known that the testis is an organ with high transcriptome diversity, AS is still understudied in human spermatogenesis. Making use of a powerful bioinformatic technique, the DTU analysis, we were able to study, for the first time, transcriptome changes at isoform resolution during human spermatogenesis. Although several studies have observed discontinuous patterns of transcription throughout murine and human spermatogenesis (Jan et al, 2017; Vara et al, 2019), in our study, we further characterized the ongoing changes in transcript levels during human spermatogenesis by identifying between 1,062 and 2,153 genes whose transcripts were alternatively spliced in different germ cell types. Our results indicate that alternative splicing extends the transcriptome diversity in germ cells, which already present high transcriptional activity, as we found that alternative splicing events are more prevalent between the pre-meiotic and meiotic germ cell types. As we identified more alternatively spliced genes than changes in gene expression between the round spermatid arrest and control samples, we hypothesize that in the final stage of spermiogenesis, transcriptome diversity arises primarily from alternative splicing rather than by changes in gene transcription. In line with this idea are studies in mice showing that genes required for spermiogenesis are already expressed at the beginning of meiosis (da Cruz et al, 2016) and that transcription in elongated spermatids is decreased because of the highly compacted chromatin structure (Sassone-Corsi, 2002). Even in the absence of transcriptional activity in the nucleus, stored unprocessed transcripts can maintain translational activity in late stages of germ cell differentiation (Wang et al, 2020). Furthermore, our study demonstrates that alternative splicing is uncoupled from the level of gene expression during human spermatogenesis, as only a minority of genes (3–8%) were both differentially expressed and differentially spliced at each respective germ cell stage. Data on the comparison of DEG and DTU genes in healthy and diseased muscle and brain tissues also revealed a small overlap (Dick et al, 2020;

Marques-Coelho et al, 2021; Solovyeva et al, 2021). Whether this is true for other tissues remains to be elucidated. Interestingly, we found that DEGs were predominantly associated with germ cell–specific processes, whereas DTU genes were involved in more general biological processes, suggesting that during human spermatogenesis, these functions are predominantly regulated at transcriptional and post-transcriptional levels, respectively. We suggest that general processes are uncoupled from the level of gene expression as these need to be maintained even in transcriptionally silent cells such as later germ cells.

By looking more closely into four DTU genes, we demonstrate the importance of our dataset for further research in the field of male infertility. For example, we were able to reveal that SPG and SPC samples express different protein-coding transcripts of *SOX15*, something that would have been overlooked by conventional DEG analysis. Our findings demonstrate the importance of understanding which gene products with potentially different functionality are produced by AS as it has been shown that this may play a role in the etiology of several diseases (Scotti & Swanson, 2016) such as cancer (Wiesner et al, 2015; Vitting-Seerup & Sandelin 2017). How alterations in alternatively spliced transcript expression play a role in the pathology of infertility remains to be assessed. We showed that some crucial spermatogenic genes such as *SYCP3* appear to be regulated at both the transcriptional and post-transcriptional levels. *SYCP3* is already expressed as an immature non-coding transcript with a retained intron in SPG samples, whereas the mature transcript is predominantly expressed in SPC samples, suggesting intron retention is a mechanism to produce transcripts required for later differentiation steps. Our hypothesis is supported by a study in mice that showed intron retention ensures timely and stage-dependent gene expression during spermatogenesis (Naro et al, 2017). In humans, a previous study indicated that spermatogonia already express genes required for meiosis (Jan et al, 2017), but the mechanism behind this observation was not addressed. Our data strongly highlight the need to further analyze the splicing machinery in human germ cells.

Although we can report on germ cell–specific transcriptome patterns that include non-coding RNAs and other RNA biotypes not covered by existing scRNA-seq studies on the human testis, we cannot address rRNAs because of rRNA depletion before total RNA sequencing. Moreover, we included samples based on careful histological examination and homogeneous histological phenotypes rather than on underlying etiologies. Therefore, the changes in gene expression we report can be confidently traced to the presence or absence of certain germ cell types rather than, for example, underlying genetic variants. For the same reason, we cannot exclude a common effect of arrests on gene expression, especially deriving from the interplay between different cell types. In the future, it will be important to validate these findings in healthy testicular tissue and discriminate between cell-specific and arrest-specific gene expression patterns.

Our whole transcriptome analysis approach provides an unbiased evaluation of transcriptome patterns during human spermatogenesis for novel and/or germ cell–specific genes. By not only focusing on protein-coding exons but by capturing the presence of all alternative transcripts at different germ cell stages, including non-coding RNAs and splice variants, our dataset increases the

understanding of human spermatogenesis and its transcriptional regulation. Our framework ultimately helps with the interpretation of pathologic variants associated with male infertility.

# Materials and Methods

### Ethical approval

Male infertility patients included in this study underwent surgery for microdissection testicular sperm extraction (mTESE; n = 15) or to rule out a suspected malignant tumor (n = 1) at the Department of Clinical and Surgical Andrology of the Centre of Reproductive Medicine and Andrology, University Hospital of Münster. Each patient gave written informed consent (ethical approval was obtained from the Ethics Committee of the Medical Faculty of Münster and the State Medical Board no. 2008-090-f-S) and one additional testicular sample for the purpose of this study was obtained. Tissue proportions were snap-frozen or fixed in Bouin's solution.

### Patient selection

In this study, we included testicular biopsies with a homogenous histological phenotype in both testes from men showing SCO (SCO-1/M1045, SCO-2/M911, SCO-3/M1742), spermatogenic arrests at the spermatogonial (SPG-1/M1570, SPG-2/M1575, SPG-3/M1072, SPG-4/M2822), spermatocyte (SPC-1/M1369, SPC-2/M799, SPC-3/M921), and round spermatid stage (SPD-1/M2227, SPD-2/M1311, SPD-3/M1400) (Table 1). For complete histological evaluation, the interstitium of each biopsy was ranked with parameters describing the condition of the tubular wall, Leydig cells, and lumen (Table S1). We excluded patients with germ cell neoplasia and a history of cryptorchidism and acute infections. For complete representation of the spermatogenic process, samples with qualitatively and quantitatively normal spermatogenesis were included as controls (CTR) in this study (CTR-1/M1544, CTR-2/M2224, CTR-3/M2234) obtained from patients with obstructive azoospermia, for example, because of congenital bilateral absence of the *vas deferens* (CBAVD; CTR-1), anorgasmia (CTR-2) or because of suspected tumor that was not confirmed (CTR-3). Before surgery, all patients underwent physical evaluation, hormonal analysis of luteinizing hormone (LH), follicle-stimulating hormone (FSH), and testosterone (T), and semen analysis (World Health Organization, 2010). In addition to conventional karyotyping and screening for azoospermia factor (AZF) deletions, WES was performed for all patients, except for SPG-4 (who had undergone chemotherapy because of leukemia) and for CTR-3. WES data were generated within the Male Reproductive Genomics (MERGE) study as previously published (Wyrwoll et al, 2020) and were screened for variants in 230 candidate genes that have at least a limited level of evidence for being associated with male infertility according to a recent review (Houston et al, 2021). We also included a screening in the recently published genes *ADAD2, GCNA, MAJIN, MSH4, MSH5, RAD21L1, RNF212, SHOC1, STAG3, SYCP2, TERB1, TERB2, TRIM71, ATG4D, BRDT, CCDC155, CHD5, CTCFL, C11orf80, C14orf39, DDX25, EXO1, GCNA, FBXO43, FKBPL, HENMT1, HFM1, HSF2, KASH5, MAGEE2, MBOAT1, MCMDC2, MCM8, MCM9, MLH3, MOV10L1,*

*PDHA2, PIWIL2, PNLDC1, PSMC3IP, RBM5, REC8, RPL10L, SPATA22, TDRD9, TDRKH, ZFX, ZSWIM7* which are associated with non-obstructive azoospermia (Riera-Escamilla et al, 2019; Krausz et al, 2020; Schilit et al, 2020; Hardy et al, 2021; Salas-Huetos et al, 2021; Torres-Fernández et al, 2021; Wyrwoll et al, 2021). We screened for rare (minor allele frequency [MAF] in gnomAD database < 0.01), possibly pathogenic variants (stop-, frameshift-, splice-site variants, and missense variants with a CADD score > 25) with a read depth > 10x, which were detected in accordance with the reported mode of inheritance in genes associated with non-syndromic infertility.

### Histological evaluation of the human testicular biopsies

After overnight fixation in Bouin's solution, the tissues were washed in 70% ethanol, embedded in paraffin, and sectioned at 5 µm. AppiClear (Cat# A4632.2500; Applichem) was used to dewax the tissue section. The cellular composition of all testicular biopsies (n = 16) was histologically examined on two periodic acid-Schiff (PAS)-stained sections from two independent biopsies per testis. For PAS staining, the sections were first incubated with 1% PA (Cat# 1.005.240.100; Sigma-Aldrich) and then in Schiffs reagent (Cat# 1.090.330.500; Sigma-Aldrich). Cell nuclei were counterstained with Mayer's hematoxylin solution (Cat# 1.092.490.500; Sigma-Aldrich). After washing in tap water and dehydration through increasing ethanol concentrations and AppiClear, slides were closed with Merckoglas (Cat# 1.039730.001; Sigma-Aldrich). The slides were scanned using the Precipoint Viewpoint software (Precipoint). The biopsies were evaluated based on the Bergmann and Kliesch scoring method (Bergmann & Kliesch, 2010), which assigns a score from 0 to 10 to each patient according to the percentage of tubules containing elongated spermatids. Furthermore, the percentage of the seminiferous tubules with round spermatids, spermatocytes, or spermatogonia as the most advanced germ cell type was assessed and seminiferous tubules with SCO or hyalinized tubules (tubular shadows) (Table 1).

### Immunohistochemical and immunofluorescence stainings on testicular tissue sections

Immunohistochemical (IHC) and immunofluorescence (IF) stainings were performed as previously described (Di Persio et al, 2021). After rehydration, heat-induced antigen retrieval in sodium citrate buffer, pH 6.0, was performed. Incubation and washing steps were performed at room temperature unless otherwise stated.

For IHC stainings, blocking of endogenous peroxidase activity and of unspecific antibody binding was achieved using hydrogen peroxide (Cat# GH06201; Hedinger) and goat serum (Cat# G6767-100ML; Sigma-Aldrich) diluted in TBS containing bovine serum albumin (Cat# A9647-50G; Sigma-Aldrich), respectively. Primary antibodies for leucine zipper protein 4 gene (LUZP4, HPA046436, 1:50; Sigma-Aldrich), testis-specific protein Y-linked 4 (TSPY4, HPA049384, dilution 1:20; Sigma-Aldrich), and high mobility group box 4 (HMGB4, HPA035699, dilution 1:50; Sigma-Aldrich) were diluted in blocking solution and incubated overnight at 4°C. Incubation with unspecific immunoglobulin G (IgG) served as negative controls. After this, sections were incubated with goat anti-rabbit biotin-labeled secondary antibody (Cat# ab6012, dilution 1:100; Abcam) for

1 h, followed by a 45-min incubation step with streptavidin–horseradish peroxidase from *Streptomyces avidinii* (Cat# S5512; Sigma-Aldrich). Detection of the peroxidase activity was achieved by incubation with 3,30-diaminobenzidine tetrahydrochloride solution (Cat# A0596.0001; Applichem) and stopped by washing in double distilled water. Nuclei were counterstained with Mayer's hematoxylin. The sections were dehydrated with increasing ethanol concentrations, cleared with AppiClear, and mounted under a glass cover slip with Merckoglas. Digitalization of the sections was performed with the Olympus BX61VS microscope and scanner software VS-ASW-S6 (Olympus).

For IF stainings, tissues were incubated with 1M glycine (Cat# G7126-500G; Sigma-Aldrich) and with a blocking solution containing TWEEN-20 (Cat# 655205; Sigma-Aldrich) and sterilized donkey serum (Cat# LIN-END9000-100; Biozol). Primary antibodies for TSPY4 (HPA049384, dilution 1:20; Sigma-Aldrich), undifferentiated embryonic cell transcription factor 1 (UTF1, MAB4337, 1:20; Merck Millipore), and MAGE family member A4 (MAGEA4, Prof. G. C. Spagnoli, University Hospital of Basel, CH, 1:20) were diluted in blocking solution and incubated overnight at 4°C. Incubation with unspecific IgG served as negative control. The next day, sections were washed and incubated for 1 h with species-specific secondary antibodies (donkey anti-rabbit Alexa 488, Cat# 711-546-152; Jackson Immuno-Research; donkey anti-mouse Alexa 647, Cat# 715-606-150; Jackson ImmunoResearch) diluted in blocking solution. Cell nuclei were counterstained with 4,6-diamidino-2-phenylindole-dihydrochloride (DAPI, Cat# D9542-10MG, 1:1,000; Sigma-Aldrich) in TBS for 10 min. After a last washing step, slides were mounted with Vectashield Mounting Media (Cat# VEC-H-1000; Vector Laboratories). Digitalization of the sections was performed with the Olympus BX61VS microscope and scanner software VS-ASW-S6 (Olympus). After immunofluorescence analyses, TSPY4, MAGEA4, and UTF1 stained cells were quantified using Qupath 0.3.2 (Bankhead et al, 2017), as described by Di Persio et al (2021). The number of TSPY4+ cells among MAGEA4+ and UTF1+ cells was quantified in three independent patient samples with full spermatogenesis. The percentages of TSPY4+ cells per sample were calculated among 200 MAGEA4+ and UTF1+ cells, respectively.

### RNA extraction from testicular tissues

We extracted total RNA from snap-frozen testicular tissues from all biopsies using the Direct-zol RNA Microprep kit (Zymo Research) according to manufacturer's protocol. Quantity and quality of isolated RNA were evaluated using RNA ScreenTape and the TapeStation Analysis software 3.1.1 (Agilent Technologies, Inc.). All samples had intact ribosomal 18S and 21S bands. Samples with an RNA integrity number (RIN) > 3.6 were included in the analysis as for human tissues, it has been shown that samples with much lower RIN values (1 < RIN < 2) can have a sufficient number of reads and pass quality control (Suntsova et al, 2019).

### Library preparation and sequencing

Next-generation sequencing was performed by the service unit Core Facility Genomics of the medical faculty at the University of Münster. Libraries were prepared according to the NEBNext Ultra

RNA II directional Library Prep kit (New England Biolabs) after NEBNext rRNA depletion (New England Biolabs). The NextSeq HO Kit (Illumina Inc.) with 150 cycles was used for paired end sequencing on the NextSeq 500 system (Illumina Inc.) with ~400 million single reads per run.

### Data processing

We processed the raw sequence data with the Nextflow analysis pipeline nf-core/rnaseq 2.0 (Ewels et al, 2020) and annotated the transcripts with GENCODE release 36 genome annotation based on the GRCh38.p13 genome reference (Frankish et al, 2019). Gene expression counts were estimated using *Salmon* (Patro et al, 2017).

### DEG analysis

All data were analyzed within the R Statistical Environment (RCoreTeam, 2020). We used DESeq2 (Love et al, 2014) for analyzing differentially expressed genes (DEGs) following the standard workflow for Salmon quantification files. DESeq2 uses a generalized linear model based on estimated size factors and dispersion to calculate the $\log_2$ fold changes for each gene (Love et al, 2014). Annotation was performed using the biomaRt R package. Normalization was performed using DESeq2 with the median of ratios method (Love et al, 2014). Genes with a total count > 10 were considered for further analysis. DEGs were calculated for each group comparison, that is, SCO versus SPG, SPG versus SPC, SPC versus SPD, and SPD versus CTR. *P*-values are calculated based on Wald test and adjusted with Benjamini–Hochberg. Genes with a false discovery rate (FDR) < 0.05 and a $\log_2$ fold change (FC) ≥ 1 were considered DEGs. Dispersion of samples was visualized using DESeq2's *PCAplot* function for the top 500 genes with a total count > 10.

To evaluate gene expression of selected genes of interest at single-cell level, we generated uniform manifold approximation and projection (UMAP) plots (McInnes et al, 2020 *Preprint*) based on our previously published dataset (Di Persio et al, 2021) using the tool Seurat (Stuart et al, 2019; Hao et al, 2021). We used the freely available loupe cell browser (v4.0.0) from 10x Genomics, Inc. to generate t-distributed stochastic neighbor embedding (t-SNE) plots of selected genes in an additional scRNA-seq dataset (Hermann et al, 2018).

### Differential transcript usage analysis

For computing differential transcript usage (DTU), we employed the R package DTUrtle (Tekath & Dugas, 2021), following the vignette workflow for human bulk RNA-seq analysis. As for the DEG analysis, we annotated the transcripts with GENCODE release 36 genome annotation. We calculated DTU genes for each group comparison (i.e., SCO versus SPG, SPG versus SPC, SPC versus SPD, and SPD versus CTR) with the *run_drimseq* function. DTUrtle conducts statistical analyses based on DRIMSeq (Nowicka & Robinson, 2016), that is, a likelihood ratio test is used on the estimated transcript proportions and precision parameter (Tekath & Dugas, 2021). To increase the statistical power of the analysis, we filtered out transcripts with low impact, that is, less than 5% usage for all

samples or a corresponding total gene expression of less than five counts for all samples before the statistical testing. Also, only genes with at least two high impact transcripts were considered. From the analysis, we obtained genes with an overall significant change in transcript usage and the corresponding transcripts that drive the change in usage in those genes (both with overall FDR < 0.05). DTUrtle conducts a conservative selection of transcripts contributing to change in isoform usage by disregarding transcripts with a potential priming bias (Tekath & Dugas, 2021).

To decrease the number of analyzed transcripts per DTU gene, a post hoc filtering was applied; that is, transcripts whose proportional expression deviated by less than 10% between samples were excluded. In this study, we decided to only include transcripts that fulfill the criterion that all samples from one group must have a higher transcript usage compared with all samples from the other group.

### Pathway analyses

Molecular functions of DEGs and DTU genes were assessed using Ingenuity Pathway Analysis (IPA; QIAGEN) and the Reactome Knowledgebase v81 (Gillespie et al, 2022). A Benjamini–Hochberg multiple testing correction *P*-value (FDR) < 0.01 was used as threshold for significant molecular functions in IPA. We selected the top 20 significant terms for molecular functions.

### cDNA synthesis and quantitative PCR analysis of testicular tissues

cDNA was synthesized from 500 ng total RNA using the iScript cDNA Synthesis Kit (Bio-Rad) according to the manufacturer's instructions. cDNA was diluted 1:3 with nuclease-free water (QIAGEN). RT-qPCR analyses were performed with PowerSYBR Green Mastermix (Life Technologies GmbH, Applied Biosystems). 1.5 $\mu$l cDNA was used for each 15 $\mu$l PCR reaction. The PCR program consisted of one cycle of 95°C for 10 min, followed by 40 cycles of 95°C for 15 s and 60°C for 1 min on a StepOnePlus machine, and results were analyzed using the StepOne software. Results for gene expression were normalized to the reference gene *GAPDH* and are plotted as $2^{-\Delta Ct}$ values (Livak & Schmittgen, 2001; Schmittgen & Livak, 2008). DTUs were calculated as the relative incidence of variants (RIV) (Camacho Londoño & Philipp, 2016) based on the relation of the specific isoform to the overall expression of its gene. Primer sequences and product sizes are summarized in Table S13.

### Statistical analysis

Statistical analysis was conducted as described in sections for DEG analysis, differential transcript usage analysis, and pathway analysis.

## Data Availability

The testicular RNA-Seq data from this publication have been deposited in the European Genome-Phenome Archive and are available under EGAS00001006135.

## Supplementary Information

## Acknowledgements

We thank Heidi Kersebom and Elke Kößer for histological evaluation of testicular tissues and Karen Schiwon for support with histological stainings. We also thank Sabine Forsthoff for excellent support in endocrinological measurements. We thank the service unit Core Facility Genomik of the medical faculty from the University of Münster for performing the next-generation sequencing. We thank Celeste Brennecka for her assistance with language editing. Schematic drawings of testicular tissues in Figs 1, 3, and 6 were created with BioRender.com. This work was funded by the German research foundation (CRU362 grants to N Neuhaus (NE 2190/3-1, NE 2190/3-2), S Laurentino (LA 4064/3-2), F Tüttelmann (TU 298/4-1, 4-2, 5-1, 5-2, 7-1), J Gromoll (GR 1547/24-2), and a pilot project to H Krenz; individual research grant to S Laurentino (LA 4064/4-1)) and by institutional funding by the CeRA. We acknowledge support from the Open Access Publication Fund of the University of Münster. The manuscript contains more specific information on the contribution of each author to the work.

## Author Contributions

LM Siebert-Kuss: formal analysis, validation, visualization, and writing—original draft, review, and editing.
H Krenz: data curation, software, formal analysis, visualization, methodology, and writing—original draft.
T Tekath: data curation, software, methodology, and writing—original draft.
M Wöste: data curation, software, methodology, and writing—original draft.
S Di Persio: formal analysis, investigation, and writing—original draft, review, and editing.
N Terwort: formal analysis, investigation, methodology, and writing—original draft.
MJ Wyrwoll: resources, data curation, formal analysis, investigation, methodology, and writing—original draft, review, and editing.
J-F Cremers: resources, data curation, investigation, and writing—review and editing.
J Wistuba: formal analysis, investigation, methodology, and writing—review and editing.
M Dugas: software, methodology, and writing—review and editing.
S Kliesch: data curation, investigation, and writing—review and editing.
S Schlatt: resources, investigation, and writing—review and editing.
F Tüttelmann: conceptualization, data curation, formal analysis, funding acquisition, and writing—original draft, review, and editing.
J Gromoll: conceptualization, funding acquisition, investigation, methodology, and writing—original draft, review, and editing.
N Neuhaus: conceptualization, formal analysis, supervision, funding acquisition, methodology, project administration, and writing—original draft, review, and editing.
S Laurentino: conceptualization, supervision, funding acquisition, investigation, visualization, methodology, project administration, and writing—original draft, review, and editing.

## Conflict of Interest Statement

The authors declare that they have no conflict of interest.

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
