## [Reviewer comments · Life Science Alliance]

Life Science Alliance

Transcriptome analyses in infertile men reveal germ cell-specific expression and splicing patterns

Lara Siebert-Kuss, Henrike Krenz, Tobias Tekath, Marius Wöste, Sara Di Persio, Nicole Terwort, Margot Wyrwoll, Jann-Frederik Cremers, Joachim Wistuba, Martin Dugas, Sabine Kliesch, Stefan Schlatt, Frank Tüttelmann, Joerg Gromoll, Nina Neuhaus, and Sandra Laurentino

DOI: <https://doi.org/10.26508/lsa.202201633>

Corresponding author(s): Sandra Laurentino, University of Münster and Nina Neuhaus, University of Münster

Review Timeline:

Submission Date:	2022-07-26
Editorial Decision:	2022-07-27
Revision Received:	2022-08-31
Editorial Decision:	2022-10-07
Revision Received:	2022-11-03
Editorial Decision:	2022-11-03
Revision Received:	2022-11-07
Accepted:	2022-11-08

Transaction Report:

Please note that the manuscript was reviewed at Review Commons and these reports were taken into account in the decision-making process at Life Science Alliance.

Manuscript number: RC-2022-01380

Corresponding author(s): Laurentino, Sandra

1. General Statements

We are grateful to the three reviewers for their careful and thoughtful reports on our manuscript. We are certain they will contribute to increase the quality of the work we are presenting and the impact of our manuscript.

The goals of this study were to assess whole transcriptome patterns and to determine the extent of alternative splicing throughout human spermatogenesis.

Through an unbiased evaluation of transcriptome patterns for novel and/or germ cell-specific genes, our study points to large transcriptional variations during human spermatogenesis, including differential gene expression and differential usage of splice variants between different spermatogenic cell stages.

The impact of our study lies in the fact that, until now, germ cell-specific whole transcriptome profiles were not clearly elucidated. Our dataset increases the understanding of transcriptional regulation during human spermatogenesis by not only focusing on protein coding exons but capturing non-coding RNAs and splice variants at different germ cell-stages. In doing this, our study highlights the need for an increased awareness of transcriptional variations during germ cell differentiation.

Peer review comments:

We are thankful to the reviewers for their overall positive comments. This study will significantly benefit from the highly thoughtful and constructive peer-review of the three reviewers. We summarize the main concerns that are common between the reviewers as following:

- The dataset has the potential for more analyses, especially on the top DEGs and genes putatively associated with the pathological conditions in the samples.

- Bold conclusions and statements should be avoided.
- An awareness on the limitations of this study needs to be pointed out.

Based on the suggestions by the reviewer, we already addressed a number of these concerns and additional experiments and analyses are ongoing. Details of these changes can be found in sections 2, 3, and 4 of this revision plan. The full comments by the reviewers and our detailed responses, as found in the revision letter, are in blue below a brief description of each item.

2. Description of the planned revisions

As requested by Reviewer #1 we will:

- Categorize the top 100 genes per group comparison regarding their (germ cell-specific) expression patterns in 3 independent scRNA-seq datasets and show the germ cell-specificity of our selected new marker genes also in the scRNA-seq dataset from Mahyari *et al.* (2021), as we already show in the Hermann *et al.* (2018) scRNA-seq dataset (figure I).
- Establish new immunohistochemical markers for specific germ cell stages.

Detailed response:

Reviewer #1 comment: (1) *Deep analysis of stage-specific genes. As the authors mention, recent scRNA-seq studies have already identified hundreds of genes exhibiting enriched expression in different male human germ cell stages. These studies have even identified genes showing preferential expression in SUBSETS of well-established germ cell stages. It is important that the present study advance the field beyond what these scRNA-seq studies have done. For example, the authors may be able to identify enriched stage-specific genes not previously identified because of the low-depth of scRNA-seq. To make this analysis comprehensive, the authors must compare not only with their own previously published scRNA-seq study, but also those from other laboratories. For at least one or two of the newly identified stage-specific markers, the authors should verify by in situ hybridization or IHC/IF analysis.*

Revision Plan

Response: We highly appreciate this comment on the importance of extending scRNA-seq data with more in-depth total RNA-seq data.

To address the point raised by the reviewer regarding expanding beyond the results of scRNA-seq studies, we will provide an extended list, as a supplementary table, of the top 100 genes per group comparison categorized according to a) genes with germ cell-specific expression in scRNA-seq datasets, b) genes that are not germ cell-specific in scRNA-seq datasets, c) genes that are not detected in scRNA-seq. We will also calculate how many of the top 100 DEGs were “germ cell stage specific” according to our criteria, as also suggested by reviewer #1 in comment (5). For this approach we will be using two more scRNA-seq datasets published by Hermann et al. (2018) and Mahyari et al. (2021). We have already assessed expression of the germ cell-specific candidate genes mentioned in our manuscript in the dataset by Hermann et al. (2018), which confirmed our findings. The corresponding preliminary figure I (below) will be included in the revised version as supplemental data.

Regarding the validation of our results also at protein level, we consider this an important step to strengthen our findings. We have ordered antibodies for spermatogonial, spermatocyte, and spermatid markers, which have partly arrived. Preliminary immunohistochemical (IHC) stainings for two spermatogonial and one round spermatid marker on testicular sections with complete spermatogenesis worked successfully. We are confident that in the following weeks, we will have completed the analyses on these markers and add the results for spermatocyte markers. Results from the IHC will be added to the revised manuscript as an additional figure.

Reviewer #1 comment: (5) Top-100 analysis. For the identification of "novel germ cell-specific marker genes," the authors focused on the top 100 DEGs from their RNA-seq analysis, and they showed three as examples. The authors should examine all of these top 100 DEGs to determine whether they follow their "germ cell stage specific" rule.

Response: As mentioned in our response to this reviewer’s comment (1), we are including a supplementary table with the top 100 genes per group comparison, where we classify the genes according to their cell-specific expression in scRNAseq datasets and according to our “germ cell stage specific” rule.

Response letter figure I (preliminary new supplementary figure): Our germ cell-specific DEGs show state-specific expression pattern also in the scRNA-seq dataset from Hermann *et al.* (2018).

In response to reviewers #1 and #3 we will:

- Provide a supplementary list of putatively misregulated genes associated with infertility and analyze their germ cell-specificity in scRNA-seq datasets.

Detailed response (underlined is what is addressed in section “3. Description of the revisions that have already been incorporated in the transferred manuscript”):

Reviewer #1 comment: (2) Misregulated genes as a result of infertility. By virtue of performing RNAseq analysis of different classes of infertility patients, the authors have a unique opportunity to identify genes MISREGULATED as a RESULT of spermatogenic defects (rather than only identifying stage-specific genes). Such "infertility misregulated" genes will be those that exhibit differential expression in a given class of infertility patient, but do NOT exhibit enriched expression at the relevant stage. Such genes can be identified in many ways, including using existing human testes scRNA-seq datasets to subtract all genes that are simply enriched at a given stage. While the authors may argue that it will be difficult to prove that such genes are misregulated in infertility patients, this is not a reasonable argument for not making such a list of such genes. To increase confidence, the misregulation of a small subset of such genes could be verified by qPCR analysis of the relevant purified germ cell subset. Even if this verification experiment is not done, a list of putative misregulated genes in different classes of infertility patients will be of great interest to the field and will illuminate candidate genes that CAUSE such cases of infertility.

Response: We agree with the reviewer that it is highly interesting to obtain a list of potential infertility-related genes. We have not done so in the original manuscript for the reasons pointed out by the reviewer. For the revised version of the manuscript however we are in the process of generating the suggested list and will analyze the cell-type specific expression in scRNA-seq datasets. The data will be added as a Supplemental table & Figure. After preparing the lists of genes we will assess to which extent it will be possible to validate our findings, e.g. in scRNA-seq datasets or by qPCR as suggested by the reviewer.

Reviewer #3 comment (1)

Major comments:

The main problem relates to the experimental strategy. Based on a couple of previous microarray studies, the authors state that "the comparison of distinct arrest phenotypes allows the identification of transcripts expressed at specific stages of germ differentiation during normal spermatogenesis". Although this is true (most probably for a high number of genes), they cannot claim that they "unveiled the transcriptional profiles of male germ cell types", or

Revision Plan

that they used this approach "to fully characterize the transcriptional profiles of human male germ cells". The genes that are up/downregulated in the different pathological conditions cannot be directly attributed to the corresponding cell types in a normal testis context. For instance, Sertoli cells may most probably express some different genes in SCO syndrome than in the presence of germ cells (the importance of the cross-talk should not be underestimated). Likewise, spermatogenic cells that are arrested at a certain stage will most probably present some aberrant gene expression. Particularly, the expression times for some genes may be altered in arrested testes (e.g. normal spermatogonia may express a subset of genes that are not being detected in arrested spermatogonia and therefore will be attributed to spermatocytes, etc.). Moreover, as different pathological conditions with spermatogenic arrest are used for each pairwise comparison, even for the same cell types that are present in both compared conditions, some of the genes that are aberrantly expressed may differ between them.

The authors must be aware of the limitations of the approach. In this regard, they must rethink their objective, and the information and conclusions they draw from this study.

Response: We agree with the reviewer and changed our conclusions to less bold statements (Lines 38-40 & 45-49 & 404-408). Indeed the approach we used for this study has some limitations, which we have now discussed in an additional paragraph (see below and lines 393-403 in the new version of the manuscript). To address the reviewer's important question regarding changes that might be related to the respective germ cell arrest, we additionally analyzed genes putatively associated with the pathological conditions in our samples, which is also in line with a comment by reviewer #1. For the revised version of the manuscript we are in the process of compiling the suggested list and will analyze the cell-type specific expression in scRNA-seq datasets. The data will be added as a Supplemental table & Figure. After preparing the lists of genes we will assess to which extent it will be possible to validate our findings.

Paragraph added to the manuscript as "Limitations of the study", lines 393-403:

"Our analysis of the testicular transcriptome has its limitations. Although we are able report on germ cell-specific transcriptome patterns that include non-coding RNAs and other RNA biotypes not covered by existing scRNA-seq studies on the human testis, we cannot address rRNAs due to rRNA depletion prior to total RNA-sequencing. Moreover, we included samples based on careful histological examination and homogeneous histological phenotypes rather than on underlying etiologies. Therefore, the changes in gene expression we report can be

confidently traced to the presence or absence of certain germ cell-types, rather than e.g. underlying genetic variants. For the same reason we cannot exclude a common effect of arrests on gene expression, especially deriving from the interplay between different cell types. In the future it will be important to validate these findings in healthy testicular tissue and discriminate between cell-specific and arrest-specific gene expression patterns.”

In response to reviewer #1 we will:

- Validate DTU genes via qPCR.

***Reviewer #1 comment:** (3) Alternative splicing. The authors should verify some examples of stage-specific AS by RT-PCR.*

Response: We designed primers specific for a set of stage-specific AS variants in order to perform this validation. We are now in the process of testing them and quantifying the expression of the different transcripts in the same samples as used for RNA-Seq using SYBR Green-based quantitative RT-PCR (qPCR). This verification process has already been successfully performed for *SYCP3*. As can be observed in the figure II below, using different sets of primers we could confirm the increased usage of *SYCP3-204* in SPG compared to SPC samples via qPCR (figure II C), which validates the original bulk RNA-seq data originally included in manuscript for gene expression and isoform usage of *SYCP3* (figure II D). We are confident that we will be able to validate other DTUs in a similarly satisfactory manner. The validation of DTUs by qPCR will be added to the manuscript as a supplementary figure.

Response letter figure II: Validation of DTU. (A) We used two primer pairs, one capable of amplifying all *SYCP3* isoforms (in the figure only *SYCP3-202* is shown as an example) while the other specifically amplify only *SYCP3-204*. The arrows indicate roughly the primer location in the transcripts. (B) Representative agarose gel electrophoresis of the amplicons obtained for each primer pair, showing a band of the expected size of an amplicon shared by all *SYCP3* isoforms (82 bp) and an amplicon specific for *SYCP3-204* (235 bp), respectively. (C & D) Comparison of the results obtained from qPCR (C) and RNA-seq (D) experiments for gene expression of *SYCP3* and the proportional isoform usage of *SYCP3-204* in SPG (n=4) and SPC samples (n=3) from the RNA-seq cohort. The isoform usage of *SYCP3-204* based on the qPCR results was calculated as the relative incidence of variants (RIV) as described by Camacho Londoño and Philipp (2016).

Revision Plan

In response to reviewer #1 we will:

- Adjust original figure 6 (now figure 5) to show CDS, UTR regions and splicing events of the transcript isoforms.

Detailed response:

Reviewer #1 comment: (4) Fig. 6 should show the CDS and UTR regions so that the reader interpret the implications of the alternative splicing event. Splicing events that disrupt reading frame should also be indicated.

Response: We agree with the reviewer and will incorporate the CDS, UTR regions and the disrupting splicing events in the figure.

In response to reviewer #2 we will:

- Extend the list of candidate genes for WES screening and analyze the link between pathogenic variants (WES) and gene and transcript expression (RNA-seq) in our samples.

Detailed response (underlined is what is addressed in section “4. Description of analyses that authors prefer not to carry out”)

Reviewer #2 comment (3): Lines 121-122: The authors have exome sequenced (WES) DNA from the patients that the analysed testicular tissue originates from. This is a great advantage, but the WES data are only screened for variants in 243 candidate genes linked to male infertility and are not fully explored. Hence, variants outside the predefined 243 genes were not considered, which seems inadequate. All variants should be called, and the most likely pathogenic variants listed. As an example, it would be interesting to know if testicular expressed genes with likely pathogenic variants appear lower expressed in these cases. Also, if some variants are predicted to affect alternative splicing, it would be interesting if this could be reflected in the expression data.

Response: Indeed, we limited our analysis of the WES data to the 243 genes that were previously described as being likely associated with male infertility according to Houston *et al.* (2021), which is based on data published until June 2020. With the very strict filtering criteria employed, we found no (likely) pathogenic variants.

Revision Plan

To address this reviewer's comment, we have now evaluated the exomes in more detail using relaxed filtering criteria (to identify additional potentially causal variants) and also investigated potential deletions. This identified *NR5A1*:c.990G>A;p.(Glu330=) in SPG-1 (M1570) and a homozygous deletion of *SYCE1* in SPC-1 (M1369) as most likely causes for the phenotypes and infertility. Furthermore, CTR-1 (M1544) had previously been diagnosed with congenital bilateral absence of vas deferens due to *CFTR* variants (c.1521_1523delCTT;p.(Phe508del)/c.2991G>C;p.(Leu997Phe). We will add this information to the final revised manuscript.

It is of note that individual samples do not drive DEGs or DTUs calling. However, if an individual sample has a mutation in a specific gene then we can investigate whether this goes along with a change in gene or transcript expression in that specific sample.

We agree with the reviewer that this is still a limited approach and, therefore, we will update and extend the list of candidate genes. To this end, we will survey the literature published since 2020 on novel genes associated with spermatogenic failure and extend the screening for these genes. We understand the reviewer's interest in a description of all variants found by WES, however we consider the exploratory exome analysis beyond the scope of this manuscript because identification of interesting variants/genes in single cases would require careful follow-up studies before any conclusions can be drawn on their pathogenicity.

More importantly, this study is intended to build a resource for the field to allow tracking of gene and isoform expression throughout spermatogenesis at high depth, which enables the field to better estimate the consequences of misregulation in gene expression and alternative splicing. Importantly, due to strict filtering criteria for our DEG and DTU analysis, the changes we observed were not driven by a single sample but were common within a group. Therefore, it is highly unlikely that the presence of any variants would have any significant influence on DEG or DTU calling, which was the primary output of this study.

In response to reviewer #2 we will:

- Add available information from other studies regarding the overlap of DEGs and DTU genes into the discussion of the revised manuscript.

Detailed response (underlined is what is addressed in section “3. Description of the revisions that have already been incorporated in the transferred manuscript”):

Reviewer #2 comment (5): The authors analyse the differential transcript usage (DTU), but it is not clear whether all exons had equal coverage in the data. E.g. were the RNAseq coverage equally well at the 5' and 3' exons, at the 5' and the middle exon, etc. Because there is a bias

toward more alternative splicing in the 5' exons it is important to show that especially the 5'exons are well-covered in the sequencing data. Furthermore, how does the 8% DTU compare with previous studies and with somatic tissues? Is this, in fact, a high number?

Response: Indeed this is an important point to consider. However, the tool that was used for the detection of DTUs, DTUrtle (Tekath and Dugas, 2021), is not affected by priming bias and takes this into consideration. Transcripts showing a priming bias do not pass the filtering of the tool and therefore are not detected as true DTUs. We have added this information to the manuscript:

Lines 197-198: "DTUrtle conducts a conservative selection of transcripts contributing to change in isoform usage by disregarding transcripts with a potential priming bias (Tekath and Dugas, 2021)."

Reviewer #2 raises an important and interesting point when questioning whether 8% is „in fact, a high number“. To the best of our knowledge, DTU analysis has not been comprehensively investigated to this extent in other tissues. Two studies reported DTU analysis on the brain focused on the comparison between healthy and diseased states and found ~2 % and 3 % overlap between DEGs and DTU genes (Dick *et al.*, 2020; Marques-Coelho *et al.*, 2021). We will add the available information to the discussion in the revised manuscript.

In response to reviewer #2 we will:

- Compare the results by IPA and Reactome and analyze the link to potential infertility-related genes.

Detailed response:

Reviewer #2 comment (6): *This reviewer does not find the enrichment analysis in figure 5 particularly informative. Many of the terms are very broad (e.g. spermatogenesis) and appear redundant (e.g. translation, expression of protein, and synthesis of protein) due to the directed acyclic nature of the GO terms. Furthermore, why was Molecular Function chosen as the only GO term to test for enrichment? Many other gene-set libraries like bioCarta, Reactome, and WikiPathways exist.*

Response: We chose IPA analysis as it is based on curated lists of genes and outputs molecular pathways, which we found particularly informative. We do agree thought that it is interesting to show the output of different tools and compare them, therefore we have

performed Reactome analysis. To make full use out of the two GO term analyses, we will compare the results obtained by IPA and Reactome also with our analysis on potential infertility-related genes (pointed out in the response to reviewer #1 comment (2)) and will add the analyses to the revised version of the manuscript.

In response to reviewer #2 we will:

- Add a supplementary table containing information on the somatic interstitial cells for each sample.

Detailed response:

***Reviewer #2 comment (8):** Lines 142-147 and Table 1: As written, it appears that the authors did not pay attention to the somatic interstitial cells, (e.g. Leydig, infiltrating lymphocytes, and peritubular cells) in the testis. This is important as bulk RNAseq was used to analyse the transcriptome of the tissue. Please clarify.*

Response: Indeed, information on the somatic cell compartment for each sample is informative. Careful histological examination is conducted within the clinical routine for each histological biopsy. This analysis includes assessment of Leydig cells, the condition of the tubular wall and the interstitial compartment (including potential infiltration of lymphocytes). We will collect and analyse this data for each of our patients. The retrieved data will be added in form of a Supplementary table and, if necessary, the data will be interpreted in the discussion of the revised manuscript.

3. Description of the revisions that have already been incorporated in the transferred manuscript

As requested by reviewer #1:

- We added information on lower RIN values in lines 150-153.

Detailed response:

***Reviewer #1 comment:** (6) RNA quality for RNA-seq. In the M&M, the authors said: "Samples with an RNA integrity number (RIN) >3.6 were included in the analysis." (Pg 7, Lines 153-154).*

Revision Plan

Given that RIN>8 is normally regarded as good quality of RNA, RIN>3.6 indicates that RNA quality was low, which might influence the authors findings.

Response: The reviewer correctly points out that the RIN values are lower than what is generally used for RNAseq analysis. In our experience, it is difficult to obtain high RIN values from human testicular biopsies. This is not due to any problem with RNA isolation or handling, but seems to be exclusive of this kind of biological sample. However, we are convinced this does not have a detrimental effect on the data obtained. This is supported by literature showing that low RIN values are not of concern if a sample passes quality controls after sequencing (Suntsova et al., 2019). We performed comprehensive quality control analysis on our samples including sufficient quality for raw reads, read alignments, quantification (5'-3' bias, biotypes, low-counts), and reproducibility among samples, and could confirm that indeed all our samples passed this quality control. To clarify this we also added this information in the revised manuscript (see below lines 150-153).

Lines 150-153: "Samples with an RNA integrity number (RIN) >3.6 were included in the analysis, as for human tissues it has been shown that samples with low RIN values (1< RIN <2) already have a sufficient number of reads and pass quality control (Suntsova *et al.*, 2019)."

As requested by reviewer #2 and #3:

- We rephrased sentences into less bold statements.

Detailed response (underlined is what was addressed in section "2. Description of the planned revisions" and in another point in "3. Description of the revisions that have already been incorporated in the transferred manuscript"):

Reviewer #2 comment (1): *Lines 39-40+87: the authors state that "By generating the most complete human testicular germ cell transcriptome to date, we...". This is a bold statement since the study does not use scRNAseq and several of scRNAseq studies of human spermatogenesis already have been published. This reviewer suggests refraining from using such bold statements and hence rephrasing these sentences. Similarly stating that "sophisticated bioinformatic analyses" (line 89) were used, is also a bit overstated, when standard pipelines/R-packages in fact were used.*

Response: In line with the reviewer's comment, we have rephrased the indicated lines:

Revision Plan

Lines 38-40 “By generating germ cell-specific whole transcriptome profiles we unravel gene expression patterns including alternative splicing during human spermatogenesis.”

Lines 85-89 “In this study, we aimed at generating whole transcriptome profiles of human testicular germ cells. Combining the advantages of scRNA-seq data and total RNA-seq of distinct pathological phenotypes, and using state-of-the-art bioinformatic analyses (...).”

Reviewer #3 comment (1)

Major comments:

The main problem relates to the experimental strategy. Based on a couple of previous microarray studies, the authors state that "the comparison of distinct arrest phenotypes allows the identification of transcripts expressed at specific stages of germ differentiation during normal spermatogenesis". Although this is true (most probably for a high number of genes), they cannot claim that they "unveiled the transcriptional profiles of male germ cell types", or that they used this approach "to fully characterize the transcriptional profiles of human male germ cells". The genes that are up/downregulated in the different pathological conditions cannot be directly attributed to the corresponding cell types in a normal testis context. For instance, Sertoli cells may most probably express some different genes in SCO syndrome than in the presence of germ cells (the importance of the cross-talk should not be underestimated). Likewise, spermatogenic cells that are arrested at a certain stage will most probably present some aberrant gene expression. Particularly, the expression times for some genes may be altered in arrested testes (e.g. normal spermatogonia may express a subset of genes that are not being detected in arrested spermatogonia and therefore will be attributed to spermatocytes, etc.). Moreover, as different pathological conditions with spermatogenic arrest are used for each pairwise comparison, even for the same cell types that are present in both compared conditions, some of the genes that are aberrantly expressed may differ between them.

The authors must be aware of the limitations of the approach. In this regard, they must rethink their objective, and the information and conclusions they draw from this study.

Response: We agree with the reviewer and changed our conclusions to less bold statements (Lines 38-40 & 45-49 & 404-408). Indeed the approach we used for this study has some limitations, which we have now discussed in an additional paragraph (see below and lines 393-403 in the new version of the manuscript). To address the reviewer's important question regarding changes that might be related to the respective germ cell arrest, we additionally

Revision Plan

analyzed genes putatively associated with the pathological conditions in our samples, which is also in line with a comment by reviewer #1. For the revised version of the manuscript we are in the process of compiling the suggested list and will analyze the cell-type specific expression in scRNA-seq datasets. The data will be added as a Supplemental table & Figure. After preparing the lists of genes we will assess to which extent it will be possible to validate our findings.

Paragraph added to the manuscript as “Limitations of the study”, lines 393-403:

“Our analysis of the testicular transcriptome has its limitations. Although we are able report on germ cell-specific transcriptome patterns that include non-coding RNAs and other RNA biotypes not covered by existing scRNA-seq studies on the human testis, we cannot address rRNAs due to rRNA depletion prior to total RNA-sequencing. Moreover, we included samples based on careful histological examination and homogeneous histological phenotypes rather than on underlying etiologies. Therefore, the changes in gene expression we report can be confidently traced to the presence or absence of certain germ cell-types, rather than e.g. underlying genetic variants. For the same reason we cannot exclude a common effect of arrests on gene expression, especially deriving from the interplay between different cell types. In the future it will be important to validate these findings in healthy testicular tissue and discriminate between cell-specific and arrest-specific gene expression patterns.”

As requested by reviewer #2:

- We replaced “normal” with “control (CTR)” throughout the manuscript.

Detailed response:

***Reviewer #2 comment (2):** Line 110: I suggest replacing "normal" with "complete" or similar. Normal indicates that you know what is normal at a population scale and in the case of humans, "normal" would likely represent suboptimal spermatogenesis.*

Response: We have replaced “normal” with “control (CTR)” throughout the manuscript and in the figures and tables.

As requested by reviewer #2 and #3:

Revision Plan

- We added a paragraph about “limitations of the study”, which addresses the rRNA depletion and changes that might be related to the respective germ cell arrest.

Detailed response (underlined is what was addressed in section “2. Description of the planned revisions” and in another point in “3. Description of the revisions that have already been incorporated in the transferred manuscript”):

Reviewer #2 comment (4): *Line 159: The authors used rRNA depletion. Did this affect the transcriptome?*

Response: Ribosomal RNA (rRNA) usually constitutes over 90% of total RNA in the cell. Therefore it is a normal procedure in RNA-seq to deplete rRNA or to select only Poly-A-tailed RNA. As we were also interested in non-Poly-A RNA, rRNA depletion is the method of choice to capture RNAs undiluted by extensive rRNA. However, we are aware that our dataset does not allow to draw conclusions about rRNA and we therefore now mention this point not only in the materials and methods, but also in the newly added “limitations of the study” paragraph to clarify this point, specifically in lines 393-396 (below).

Lines 393-396: “Our analysis of the testicular transcriptome has its limitations. Although we can report on germ cell-specific transcriptome patterns that include non-coding RNAs and other RNA biotypes not covered by existing scRNA-seq studies on the human testis, we cannot address rRNAs due to rRNA depletion prior to total RNA-sequencing.”

Reviewer #3 comment (1)

Major comments:

The main problem relates to the experimental strategy. Based on a couple of previous microarray studies, the authors state that "the comparison of distinct arrest phenotypes allows the identification of transcripts expressed at specific stages of germ differentiation during normal spermatogenesis". Although this is true (most probably for a high number of genes), they cannot claim that they "unveiled the transcriptional profiles of male germ cell types", or that they used this approach "to fully characterize the transcriptional profiles of human male germ cells". The genes that are up/downregulated in the different pathological conditions cannot be directly attributed to the corresponding cell types in a normal testis context. For instance, Sertoli cells may most probably express some different genes in SCO syndrome

Revision Plan

than in the presence of germ cells (the importance of the cross-talk should not be underestimated). Likewise, spermatogenic cells that are arrested at a certain stage will most probably present some aberrant gene expression. Particularly, the expression times for some genes may be altered in arrested testes (e.g. normal spermatogonia may express a subset of genes that are not being detected in arrested spermatogonia and therefore will be attributed to spermatocytes, etc.). Moreover, as different pathological conditions with spermatogenic arrest are used for each pairwise comparison, even for the same cell types that are present in both compared conditions, some of the genes that are aberrantly expressed may differ between them.

The authors must be aware of the limitations of the approach. In this regard, they must rethink their objective, and the information and conclusions they draw from this study.

Response: We agree with the reviewer and changed our conclusions to less bold statements (Lines 38-40 & 45-49 & 404-408). Indeed the approach we used for this study has some limitations, which we have now discussed in an additional paragraph (see below and lines 393-403 in the new version of the manuscript). To address the reviewer's important question regarding changes that might be related to the respective germ cell arrest, we additionally analyzed genes putatively associated with the pathological conditions in our samples, which is also in line with a comment by reviewer #1. For the revised version of the manuscript we are in the process of compiling the suggested list and will analyze the cell-type specific expression in scRNA-seq datasets. The data will be added as a Supplemental table & Figure. After preparing the lists of genes we will assess to which extent it will be possible to validate our findings.

Paragraph added to the manuscript as "Limitations of the study", lines 393-403:

"Our analysis of the testicular transcriptome has its limitations. Although we are able report on germ cell-specific transcriptome patterns that include non-coding RNAs and other RNA biotypes not covered by existing scRNA-seq studies on the human testis, we cannot address rRNAs due to rRNA depletion prior to total RNA-sequencing. Moreover, we included samples based on careful histological examination and homogeneous histological phenotypes rather than on underlying etiologies. Therefore, the changes in gene expression we report can be confidently traced to the presence or absence of certain germ cell-types, rather than e.g. underlying genetic variants. For the same reason we cannot exclude a common effect of arrests on gene expression, especially deriving from the interplay between different cell types. In the future it

Revision Plan

will be important to validate these findings in healthy testicular tissue and discriminate between cell-specific and arrest-specific gene expression patterns.”

As requested by reviewer #2:

- We clarified in the manuscript that DTU analysis is not affected by priming bias.

Detailed response (underlined is what was addressed in section “2. Description of the planned revisions”):

***Reviewer #2 comment (5):** The authors analyse the differential transcript usage (DTU), but it is not clear whether all exons had equal coverage in the data. E.g. were the RNAseq coverage equally well at the 5' and 3' exons, at the 5' and the middle exon, etc. Because there is a bias toward more alternative splicing in the 5' exons it is important to show that especially the 5' exons are well-covered in the sequencing data. Furthermore, how does the 8% DTU compare with previous studies and with somatic tissues? Is this, in fact, a high number?*

Response: Indeed this is an important point to consider. However, the tool that was used for the detection of DTUs, DTUrtle (Tekath and Dugas, 2021), is not affected by priming bias and takes this into consideration. Transcripts showing a priming bias do not pass the filtering of the tool and therefore are not detected as true DTUs. We have added this information to the manuscript:

Lines 197-198: “DTUrtle conducts a conservative selection of transcripts contributing to change in isoform usage by disregarding transcripts with a potential priming bias (Tekath and Dugas, 2021).”

Reviewer #2 raises an important and interesting point when questioning whether 8% is „in fact, a high number“. To the best of our knowledge, DTU analysis has not been comprehensively investigated to this extent in other tissues. Two studies reported DTU analysis on the brain focused on the comparison between healthy and diseased states and found ~2 % and 3 % overlap between DEGs and DTU genes (Dick et al., 2020; Marques-Coelho et al., 2021). We will add the available information to the discussion in the revised manuscript.

As requested by reviewer #2 and #3:

- We rephrased original lines 49-51 (now 48-49) and original lines 414-418 (now 405-408).

Revision Plan

Detailed response:

Reviewer #2 comment (7): *Lines 49-51: This reviewer does not agree that the identification of new genetic pathogenic variants causing infertility is hindered by a lack of transcriptome data on human spermatogenesis. However, it indeed helps in understanding the functional consequences of a pathogenic variant.*

Response: We understand the reviewer's concern about the phrasing and in fact agree wholeheartedly with the reviewer's view that our dataset helps in understanding the functional consequences of a pathogenic variant. We took the liberty of using this sentence in the manuscript and thank the reviewer for helping us clarify the significance of this study.

Lines 48-49: "In this regard germ cell-specific whole transcriptome profiles would help in understanding the functional consequences of a pathogenic variant."

Reviewer #2 comment (11): *The last para of the discussion ends by stating that the data generated in the study will improve "the genetic diagnosis of male infertility". This reviewer does not understand why the authors focus on genetic diagnosis. The study increases our understanding of human spermatogenesis and does not need to be justified by potential improvements in the genetic diagnosis of infertile men.*

Response: Again, we agree with the reviewer's opinion on this point and made the appropriate changes accordingly.

Original Lines 414-418, now 405-408: "By not only focusing on protein coding exons but capturing the presence of all alternative transcripts at different germ cell-stages, including non-coding RNAs and splice variants, our dataset increases the understanding of human spermatogenesis and its transcriptional regulation."

As requested by reviewer #2:

- We combined Figures 2 and 3 into a new preliminary Figure 2 (uploaded separately).

Detailed response:

Reviewer #2 comment (9): *The heatmaps in fig 2 are not very informative and the information given in Figures 2 and 3 is quite similar. I suggest somehow combining Figures 2 and 3.*

Revision Plan

Response: We have combined Figures 2 and 3 into a new preliminary Figure 2 (uploaded separately).

As requested by reviewer #2:

- We streamlined the discussion and deleted original lines 340-7 and 352-8.

Detailed response:

Reviewer #2 comment (10): Parts of the discussion appear to be a repetition of the introduction (e.g. lines 340-7) and results (e.g. lines 348-357), which should be minimised.

Response: We have streamlined the text in accordance with this comment. Original lines 340-7 and 352-8 were deleted.

As requested by reviewer #3:

- We clarified the assessment of cellularities of each sample in the revised manuscript.

Detailed response:

Reviewer #3 comment (2): Minor comments:

-Table 1, and Fig 1B: The term SCO (Sertoli cell-only) is used instead of Sertoli cells. The percentages of spermatogonia and Sertoli cells for the normal tubules are missing.

Response: We would like to clarify that the percentages/cellularities given in Table 1 and Fig. 1B refer to the most advanced state of the tubules, therefore SCO does not correspond to the number of Sertoli cells per tubule but to the tubules containing only Sertoli cells. Moreover, in SCO-classified samples, there were no tubules containing germ cells. In contrast, our histological examination of samples with complete spermatogenesis showed that in 12% of the tubules the most advanced cell states are round spermatids or spermatocytes. Thus there are no spermatogonial arrested and Sertoli cell-only tubules in the samples with full spermatogenesis. To clarify this point, we revised the description of table 1 and figure 1 of the manuscript.

4. Description of analyses that authors prefer not to carry out

Reviewer #2 requested that we:

- Call all variants and list the most likely pathogenic ones per patient.

We will not call all variants because the identification of interesting variants/genes requires careful follow-up studies before meaningful conclusions can be made and because the strict filtering criteria for our DEG and DTU analysis leads to the detection of changes common within a group rather than due the presence of any variants. Please see our detailed explanation below.

Detailed response (underlined is what was addressed in section “2. Description of the planned revisions”):

Reviewer #2 comment (3): Lines 121-122: The authors have exome sequenced (WES) DNA from the patients that the analysed testicular tissue originates from. This is a great advantage, but the WES data are only screened for variants in 243 candidate genes linked to male infertility and are not fully explored. Hence, variants outside the predefined 243 genes were not considered, which seems inadequate. All variants should be called, and the most likely pathogenic variants listed. As an example, it would be interesting to know if testicular expressed genes with likely pathogenic variants appear lower expressed in these cases. Also, if some variants are predicted to affect alternative splicing, it would be interesting if this could be reflected in the expression data.

Response: Indeed, we limited our analysis of the WES data to the 243 genes that were previously described as being likely associated with male infertility according to Houston et al. (2021), which is based on data published until June 2020. With the very strict filtering criteria employed, we found no (likely) pathogenic variants.

To address this reviewer’s comment, we have now evaluated the exomes in more detail using relaxed filtering criteria (to identify additional potentially causal variants) and also investigated potential deletions. This identified NR5A1:c.990G>A;p.(Glu330=) in SPG-1 (M1570) and a homozygous deletion of SYCE1 in SPC-1 (M1369) as most likely causes for the phenotypes and infertility. Furthermore, CTR-1 (M1544) had previously been diagnosed with congenital bilateral absence of vas deferens due to CFTR variants (c.1521_1523delCTT;p.(Phe508del)/c.2991G>C;p.(Leu997Phe). We will add this information to the final revised manuscript.

Revision Plan

It is of note that individual samples do not drive DEGs or DTUs calling. However, if an individual sample has a mutation in a specific gene then we can investigate whether this goes along with a change in gene or transcript expression in that specific sample.

We agree with the reviewer that this is still a limited approach and, therefore, we will update and extend the list of candidate genes. To this end, we will survey the literature published since 2020 on novel genes associated with spermatogenic failure and extend the screening for these genes. We understand the reviewer's interest in a description of all variants found by WES, however we consider the exploratory exome analysis beyond the scope of this manuscript because identification of interesting variants/genes in single cases would require careful follow-up studies before any conclusions can be drawn on their pathogenicity.

More importantly, this study is intended to build a resource for the field to allow tracking of gene and isoform expression throughout spermatogenesis at high depth, which enables the field to better estimate the consequences of misregulation in gene expression and alternative splicing. Importantly, due to strict filtering criteria for our DEG and DTU analysis, the changes we observed were not driven by a single sample but were common within a group. Therefore, it is highly unlikely that the presence of any variants would have any significant influence on DEG or DTU calling, which was the primary output of this study.

July 27, 2022

Re: Life Science Alliance manuscript #LSA-2022-01633-T

Dr. Sandra Laurentino
University of Münster
Centre of Reproductive Medicine and Andrology, Institute of Reproductive and Regenerative Biology
Domagkstr. 11
Münster, NRW 48149
Germany

Dear Dr. Laurentino,

Thank you for submitting your manuscript entitled "Stage-specific gene and transcript dynamics in human male germ cells" to Life Science Alliance. We invite you to re-submit the manuscript, revised according your Revision Plan.

Thank you for this interesting contribution to Life Science Alliance. We are looking forward to receiving your revised manuscript.

Sincerely,

B. MANUSCRIPT ORGANIZATION AND FORMATTING:

Dear Dr. Sawey,

We would like to thank you for your interest in our manuscript, originally titled "Stage-specific gene and transcript dynamics in human male germ cells". We appreciate the highly constructive comments and suggestions from the three reviewers, which we addressed by performing additional experiments and analyses that can now be seen in the revised manuscript and in the following point-by-point response. To reflect more accurately the contents of this manuscript we have also updated its title to "Exploring gene expression and isoform usage during human spermatogenesis".

The revised sections of the manuscript are highlighted in yellow. In this point-by-point response the original comments by the reviewers are in shaded italic text while our responses are in plain text.

Point-by-point response to the reviewers' comments

Reviewer #1 (Evidence, reproducibility and clarity (Required)):

In this study, Siebert-Kuss et al. performed RNA-seq analysis of whole biopsies from infertile patients with an arrest at different stages of male human germ cell development. This allowed them to identify genes likely enriched in different germ cell types (using a pairwise comparison approach of adjacent stages). This verified results from recent scRNA-seq studies, and may provide a more complete list of marker genes than these past studies given that RNA-seq analysis provides much deeper reads than scRNA-seq analysis. The most novel aspect of their study is the DTU analysis, as it identified putative stage-specific alternative spliced transcripts.

This study is unique in that it makes use of difficult-to-find infertility patients with defects in specific stages of spermatogenesis. This study is a useful resource for the field and it draws some interesting conclusions. However, the authors fail to make full use of their RNAseq datasets, as described below. Thus, the potential of this study to move the field significantly forward is, at present, not fully realized. Below are specific suggestions:

Response: We thank the reviewer for the appreciation of our work. We have addressed the highly constructive comments below.

Reviewer #1 comment: *(1) Deep analysis of stage-specific genes. As the authors mention, recent scRNA-seq studies have already identified hundreds of genes exhibiting enriched expression in different male human germ cell stages. These studies have even identified genes showing preferential expression in SUBSETS of well-established germ cell stages. It is important that the present study advance the field beyond what these scRNA-seq studies have done. For example, the authors may be able to identify enriched stage-specific genes not previously identified because of the low-depth of scRNA-seq. To make this analysis comprehensive, the authors must compare not only with their own previously published scRNA-seq study, but also those from other laboratories. For at least one or two of the newly identified stage-specific markers, the authors should verify by in situ hybridization or IHC/IF analysis.*

Response: We appreciate this comment expanded and deepened our analyses. This included expanding the analysis to the top 120 DEGs per group comparison and to study their expression not

only in our own scRNA-seq dataset but also in an additional previously published dataset (Hermann et al, 2018), which also studied testicular cells from patients with full spermatogenesis. In the new supplementary figure S1 we plotted all top 120 DEGs and found that $85 \pm 9\%$ displayed germ cell-type specific expression. Moreover, we listed in a new supplementary table S7 all genes with insufficient or absent expression in the scRNA-seq datasets, as well as genes that are expressed in different cell types. For our newly identified cell-specific markers, we demonstrate the accordance between our and Hermann et al (2018) scRNA-seq datasets in a new supplementary figure S2. As suggested by this reviewer, we also verified some of our novel markers by IHC analysis and added the results to the revised figure 3, demonstrating the germ cell type-specificity also at protein level for two spermatogonial markers (LUZP4, TSPY4) and one round spermatid marker (HMGB4). The new results were also added to the revised manuscript text (lines 157-177).

Reviewer #1 comment: (2) *Misregulated genes as a result of infertility. By virtue of performing RNAseq analysis of different classes of infertility patients, the authors have a unique opportunity to identify genes MISREGULATED as a RESULT of spermatogenic defects (rather than only identifying stage-specific genes). Such "infertility misregulated" genes will be those that exhibit differential expression in a given class of infertility patient, but do NOT exhibit enriched expression at the relevant stage. Such genes can be identified in many ways, including using existing human testes scRNA-seq datasets to subtract all genes that are simply enriched at a given stage. While the authors may argue that it will be difficult to prove that such genes are misregulated in infertility patients, this is not a reasonable argument for not making such a list of such genes. To increase confidence, the misregulation of a small subset of such genes could be verified by qPCR analysis of the relevant purified germ cell subset. Even if this verification experiment is not done, a list of putative misregulated genes in different classes of infertility patients will be of great interest to the field and will illuminate candidate genes that CAUSE such cases of infertility.*

Response: We agree with the reviewer that it is highly interesting to obtain a list of potential infertility-related genes. We have not done so in the original manuscript for the reasons pointed out by the reviewer. For the revised version of the manuscript we generated the suggested list of infertility misregulated genes by extracting all genes with elevated expression in the "lower" arrest of each group comparisons namely for the SCO, SPG, SPC, and SPD phenotypes. We identified between 63 and 1,252 infertility-related genes (Table S2 and genes indicated in blue in Tables S4-S6) and performed Reactome analysis on the respective gene lists. Moreover, we analyzed the cell type-specific expression for the 50 most severely differentially expressed genes (according to \log_2FC) in our scRNA-seq dataset. The analysis revealed 4 particularly interesting genes that were upregulated in SCO (*RWDD2A*, *CCDC183*, *CNNM1*) or SPD (*SERF1B*) compared to SPG and CTR samples, respectively, but that according to scRNA-seq displayed a germ cell-specific or meiotic-specific expression pattern, respectively (Supplementary Figure S6). Accordingly, we compared the exon expression of these 4 candidate genes for male infertility with bulk tissue expression in the GTEx portal (supplementary Figure S7). We implemented the analyses in a new results paragraph "Identification of putatively misregulated genes in male infertility" (lines 241-263) and discussed our results (lines 275-281) in the revised manuscript.

Reviewer #1 comment: (3) *Alternative splicing. The authors should verify some examples of stage-specific AS by RT-PCR.*

Response: We designed primers for a gene with differential transcript usage (DTU gene) that is not a differentially expressed gene (DEG) between SPG and SPC samples (*SOX15*) as well as for one DTU gene that is also a DEG (*SYCP3*) in the same group comparison. We have analyzed *SOX15* instead of the example that was originally included in the manuscript *ACTL6A* (similarly a DTU in two protein coding transcripts but not a DEG) as the latter could not be validated by qPCR because the different transcripts could not be sufficiently distinguished using this technique. We could confirm the decreased usage of *SOX15-201* as well as increased usage of *SYCP3-204* in SPG compared to SPC samples via qPCR, validating the bulk RNA-seq findings. The validation of DTUs by qPCR was added to the manuscript as supplementary figure S5 and in lines 213-220, 233-235 and 506-518 of the revised manuscript.

Reviewer #1 comment: (4) Fig. 6 should show the CDS and UTR regions so that the reader interpret the implications of the alternative splicing event. Splicing events that disrupt reading frame should also be indicated.

Response: We adjusted the schematic transcripts in the revised figure 6 to include exons/CDS, UTR regions and introns. We also added the genomic locations and color coded the transcripts according to their biotype.

Reviewer #1 comment: (5) Top-100 analysis. For the identification of "novel germ cell-specific marker genes," the authors focused on the top 100 DEGs from their RNA-seq analysis, and they showed three as examples. The authors should examine all of these top 100 DEGs to determine whether they follow their "germ cell stage specific" rule.

Response: As mentioned in our response to this reviewer's comment (1), we included a supplementary figure S1 with the top 120 genes per group comparison, as well as a new supplementary table S7, where we list the genes with low or absent expression in the scRNA-seq datasets and genes not fulfilling the "germ cell stage specific" rule (lines 157-167).

Reviewer #1 comment: (6) RNA quality for RNA-seq. In the M&M, the authors said: "Samples with an RNA integrity number (RIN) >3.6 were included in the analysis." (Pg 7, Lines 153-154). Given that RIN>8 is normally regarded as good quality of RNA, RIN>3.6 indicates that RNA quality was low, which might influence the authors findings.

Response: The reviewer correctly points out that the RIN values are lower than what is generally used for RNAseq analysis. In our experience, it is difficult to obtain high RIN values from human testicular biopsies. This is not due to any problem with RNA isolation or handling, but seems to be exclusive of this kind of biological sample. However, we are convinced this does not have a detrimental effect on the data obtained. This is supported by literature showing that low RIN values are not of concern if a sample passes quality controls after sequencing (Suntsova et al, 2019). We performed comprehensive quality control analysis on our samples including sufficient quality for raw reads, read alignments, quantification (5'-3' bias, biotypes, low-counts), and reproducibility among samples, and could confirm that indeed all our samples passed this quality control. To clarify this we also added this information in the revised manuscript (lines 442-445).

Reviewer #2 (Evidence, reproducibility and clarity (Required)):

Summary

In the manuscript by Siebert-Kuss et al., the authors investigate the transcriptome of human spermatogenesis by bulk RNA sequencing (RNAseq) of testicular tissues with different content of germ cells. Input tissues subjected to bulk RNAseq include histologically defined testis tissues showing Sertoli Cell-Only (n=3), arrests at the spermatogonial (n=4), spermatocyte (n=3), and round spermatid (n=3) stages as well as "normal" testicular tissue (n=3). The expression data is linked to published single-cell RNAseq (scRNAseq) data and the differential expression as well as the use of alternative splicing determined. The manuscript appears well-written and easy to understand and the figures and tables are clear.

Response: We appreciate the author's appraisal of the manuscript's quality.

Major comments:

Reviewer #2 comment (1): *Lines 39-40+87: the authors state that "By generating the most complete human testicular germ cell transcriptome to date, we...". This is a bold statement since the study does not use scRNAseq and several of scRNAseq studies of human spermatogenesis already have been published. This reviewer suggests refraining from using such bold statements and hence rephrasing these sentences. Similarly stating that "sophisticated bioinformatic analyses" (line 89) were used, is also a bit overstated, when standard pipelines/R-packages in fact were used.*

Response: In line with the reviewer's comment, we have deleted the original lines 39-40 and rephrased lines 84-89 in the revised manuscript.

Reviewer #2 comment (2): *Line 110: I suggest replacing "normal" with "complete" or similar. Normal indicates that you know what is normal at a population scale and in the case of humans, "normal" would likely represent suboptimal spermatogenesis.*

Response: We have replaced "normal" with "control (CTR)" throughout the manuscript and in the figures and tables.

Reviewer #2 comment (3): *Lines 121-122: The authors have exome sequenced (WES) DNA from the patients that the analysed testicular tissue originates from. This is a great advantage, but the WES data are only screened for variants in 243 candidate genes linked to male infertility and are not fully explored. Hence, variants outside the predefined 243 genes were not considered, which seems inadequate. All variants should be called, and the most likely pathogenic variants listed. As an example, it would be interesting to know if testicular expressed genes with likely pathogenic variants appear lower expressed in these cases. Also, if some variants are predicted to affect alternative splicing, it would be interesting if this could be reflected in the expression data.*

Response: Indeed, our analysis of the WES data was limited to the 243 genes that were previously described as being likely associated with male infertility according to Houston et al (2021), which is

based on data published until June 2020. With the very strict filtering criteria employed, we found no (likely) pathogenic variants. To address this reviewer's comment, we performed a literature search, screening for further genes described in the context of azoospermia, which have been published between 2020 and 2022. Based on this literature search we now included the genes *ATG4D*, *BRDT*, *CCDC155*, *CHD5*, *CTCF*, *C11orf80*, *C14orf39*, *DDX25*, *EXO1*, *GCNA*, *FBXO43*, *FKBP1*, *HENMT1*, *HFM1*, *HSF2*, *KASH5*, *MAGEE2*, *MBOAT1*, *MCMD2*, *MCM8*, *MCM9*, *MLH3*, *MOV10L1*, *PDHA2*, *PIWIL2*, *PNLDC1*, *PSMC3IP*, *RBM5*, *REC8*, *RPL10L*, *SPATA22*, *TDRD9*, *TDRKH*, *ZFX*, *ZSWIM7* to the previously screened 243 genes in our variant screening. Next, we loosened our filtering criteria and screened not only for stop gain, frameshift and splice site variants, but also for missense variants with a CADD score >25. We have included the new information in a new results sub-section "Genetic characterization of the patient cohort" (lines 106-129) and expanded the "Patient selection" sub-section in the Materials and methods (lines 382-397). In short, this approach led to the identification of a heterozygous missense variant in patient SCO-2 (M911) in *SYCP3*, which is predicted to potentially affect splicing (NM_153694.4: c.551A>C p.(Lys184Thr)). In patient SPD-1 (M2227) we identified a heterozygous missense variant in *PLK4* with a CADD score of 28.8 (NM_014264.5 c.950C>T p.(Pro317Leu)) as well as the heterozygous variant NM_021951.3 c.355-4C>T p.? in *DMRT1*, which might also have an impact on splicing. Patient SPC-1 (M1369) was identified in a parallel study to carry a homozygous deletion affecting the complete *SYCE1* gene (Wyrwoll et al, 2022). In line with this finding, we did not detect reads of *SYCE1* in SPC-1 in our total RNA-seq data. Patient SPG-1 (M1570) was also identified in a parallel study with the heterozygous synonymous variant NM_004959.5 c.990G>A p.(Glu330=) in *NR5A1*, which is also predicted to alter splicing (Wyrwoll et al, 2022). Control patient CTR-1 (M1544) had been previously diagnosed during routine genetic diagnostics with the *CFTR* variants c.1521_1523delCTT p.(Phe508del) and c.2991G>C p.(Leu997Phe), which represent the cause for congenital absence of vas deferens in this man. We also analyzed the effect of variants with a predicted effect on splicing in the RNA-seq data. We revealed that the heterozygous synonymous variant in *NR5A1* of patient SPG-1 led to the use of an alternative 5' splice site in the affected exon 5 (new Fig. 2). For all other variants, which were predicted to affect splicing, no alternative splice sites were identified in the total RNA-seq data of the respective patients. We understand the reviewer's interest in a description of all variants found by WES, however we consider the exploratory exome analysis beyond the scope of this manuscript because identification of interesting variants/genes in single cases would require careful follow-up studies before any conclusions can be drawn on their pathogenicity. More importantly, this study is intended to build a resource for the field to allow tracking of gene and isoform expression throughout spermatogenesis at high depth, which enables a better estimation of the consequences of misregulation in gene expression and alternative splicing. Importantly, due to strict filtering criteria for our DEG and DTU analysis, the changes we observed were not driven by a single sample but were common within a group. Therefore, it is highly unlikely that the presence of any variants would have any significant influence on DEG or DTU calling, which was the primary output of this study.

Reviewer #2 comment (4): Line 159: The authors used rRNA depletion. Did this affect the transcriptome?

Response: Ribosomal RNA (rRNA) usually constitutes over 90% of total RNA in the cell. Therefore it is a normal procedure in RNA-seq to deplete rRNA or to select only Poly-A-tailed RNA. As we were also interested in non-Poly-A RNA, rRNA depletion is the method of choice to capture RNAs undiluted by extensive rRNA. However, we are aware that our dataset does not allow to draw conclusions about

rRNA and we therefore now mention this point not only in the materials and methods, but also in a paragraph on the limitations of our study paragraph (lines 335-345).

Reviewer #2 comment (5): *The authors analyse the differential transcript usage (DTU), but it is not clear whether all exons had equal coverage in the data. E.g. were the RNAseq coverage equally well at the 5' and 3' exons, at the 5' and the middle exon, etc. Because there is a bias toward more alternative splicing in the 5' exons it is important to show that especially the 5' exons are well-covered in the sequencing data. Furthermore, how does the 8% DTU compare with previous studies and with somatic tissues? Is this, in fact, a high number?*

Response: Indeed this is an important point to consider. However, the tool that was used for the detection of DTUs, DTUrtle (Tekath and Dugas, 2021), is not affected by priming bias and takes this factor into consideration. Transcripts showing a priming bias do not pass the filtering of the tool and therefore are not detected as true DTUs. We have added this information to the manuscript in lines 492-494.

Reviewer #2 raises an important and interesting point when questioning whether 8% is „in fact, a high number“. To the best of our knowledge, DTU analysis has not been comprehensively investigated to this extent in other tissues. Two studies reported DTU analysis on the brain focused on the comparison between healthy and diseased states and found ~2 % and 3 % overlap between DEGs and DTU genes (Dick et al, 2020; Marques-Coelho et al, 2021). We added the available information to the discussion in the revised manuscript in lines 304-310.

Reviewer #2 comment (6): *This reviewer does not find the enrichment analysis in figure 5 particularly informative. Many of the terms are very broad (e.g. spermatogenesis) and appear redundant (e.g. translation, expression of protein, and synthesis of protein) due to the directed acyclic nature of the GO terms. Furthermore, why was Molecular Function chosen as the only GO term to test for enrichment? Many other gene-set libraries like bioCarta, Reactome, and WikiPathways exist.*

Response: We chose IPA analysis as it is based on curated lists of genes and outputs molecular pathways, which we found particularly informative. We do agree though that it is interesting to show the output of different tools and compare them, and therefore we additionally performed Reactome analysis. To make full use out of the two GO term analyses, we compared the results obtained by IPA and Reactome for the DEGs and DTU genes of each group comparison. We added the analyses to the revised version of the manuscript in the form of Supplementary Figures S3 and S4 and in lines 204-207. Moreover, we analyzed putative infertility-related genes via the Reactome database and included this as a new supplementary figure S6 as well as in a new results paragraph (pointed out also in response to reviewer #1 comment (2)) in lines 241-263 and discussed in lines 275-281.

Minor comments

Reviewer #2 comment (7): *Lines 49-51: This reviewer does not agree that the identification of new genetic pathogenic variants causing infertility is hindered by a lack of transcriptome data on human spermatogenesis. However, it indeed helps in understanding the functional consequences of a pathogenic variant.*

Response: We understand the reviewer's concern about the phrasing and in fact agree wholeheartedly with the reviewer's view that our dataset helps in understanding the functional consequences of a pathogenic variant. We took the liberty of incorporating this sentence in the manuscript (lines 84-89) and thank the reviewer for helping us clarify the significance of this study. Based on this reviewer's comment we extended our analyses on the functional consequences of a pathogenic variant and added the results (see lines 123-129 and new figure 2).

Reviewer #2 comment (8): Lines 142-147 and Table 1: As written, it appears that the authors did not pay attention to the somatic interstitial cells, (e.g. Leydig, infiltrating lymphocytes, and peritubular cells) in the testis. This is important as bulk RNAseq was used to analyse the transcriptome of the tissue. Please clarify.

Response: Indeed, information on the somatic cell compartment for each sample is informative. Careful histological examination is conducted within the clinical routine for each histological biopsy. This analysis includes assessment of Leydig cells, the condition of the tubular wall and the interstitial compartment (including potential infiltration of lymphocytes). We collected this data for each patient and added the data in form of a Supplementary table S1 and additional information on this analysis was added to the Materials and Methods (lines 368-370).

Reviewer #2 comment (9): The heatmaps in fig 2 are not very informative and the information given in Figures 2 and 3 is quite similar. I suggest somehow combining Figures 2 and 3.

Response: We have combined Figures 2 and 3 into a new Figure 3.

Reviewer #2 comment (10): Parts of the discussion appear to be a repetition of the introduction (e.g. lines 340-7) and results (e.g. lines 348-357), which should be minimised.

Response: We have streamlined the text in accordance with this comment. Original lines 340-347 and 352-358 were deleted.

Reviewer #2 comment (11): The last para of the discussion ends by stating that the data generated in the study will improve "the genetic diagnosis of male infertility". This reviewer does not understand why the authors focus on genetic diagnosis. The study increases our understanding of human spermatogenesis and does not need to be justified by potential improvements in the genetic diagnosis of infertile men.

Response: We agree with the reviewer's opinion that this is not the main focus of the manuscript and changed the text accordingly (lines 346-352 in the revised manuscript).

Reviewer #2 (Significance (Required)):

Significance

As also listed in the manuscript, several scRNAseq studies already exist on human spermatogenesis, and as such, the bulk RNAseq approach presented does not contribute much new information. The major finding of this manuscript is that the use of alternative splicing seems uncoupled to changes in

expression during spermatogenesis. However, this could be further explored e.g., by including exon-expression data from other tissues available in the GTEx portal. Also, the link between WES and testicular expression seems unexplored.

Response: We thank the reviewer for the suggestion of further exploring the available data, which led us to enrich our data analyses. This included three major analyses that we think increased the solidity and significance of our study:

1. We identified putatively misregulated genes in our groups, as pointed out in response to reviewer #1's comment (Supplementary Figure S7), and included exon expression data from 54 tissues available in the GTEx portal (lines 241-263 and 275-281).
2. We expanded the analysis on alternative splicing and validated selected DTU genes using qPCR (Supplementary Figure S5; lines 213-220, 233-235, and 506-518).
3. We broadened the gene lists and loosened filtering criteria to analyze the WES data of each patient (lines 106-129 and 382-397). The genes where variants were found were further analyzed in the total RNA-seq data. For *NR5A1* we identified a variant that led to the use of an alternative 5' splice site in exon 5 of this gene (Figure 2), originating both the canonical transcript and an alternative transcript with a 48 bp deletion. For the other variants that were predicted to affect alternative splicing, no changes were found at RNA level.

****Referees cross-commenting****

I agree with Reviewer #1 that this study does not advance the field substantially beyond published scRNAseq studies and that the data can be explored more.

I also agree with Reviewer #3 that they overstate the results from their study.

Response: Based on the highly constructive comments from all 3 reviewers, we incorporated the suggested changes, which includes expanded analyses of our data as well as experimental work for validation of our results. We revised our discussion and conclusion accordingly. We trust that with the revised version we addressed the points mentioned here and highlight the add-on value of this combinational approach of bulk RNA-seq and scRNA-seq.

Reviewer #3 (Evidence, reproducibility and clarity (Required)):

In this ms, the authors characterize the transcriptional profiles of human male germ cells through bulk RNAseq, by comparing whole testes transcriptomes from patients with arrest at different spermatogenic stages (Sertoli cell only syndrome vs arrested at spermatogonia stage; arrested at spermatogonia vs arrested at spermatocytes stage; arrested at spermatocytes vs arrested at round spermatids; and arrested at round spermatids vs normal). They report some interesting findings concerning alternative splicing, such as the minimum overlap between differentially expressed genes and differential transcript usage.

Response: We thank reviewer #3 for the appreciation of our findings.

Reviewer #3 comment (1) Major comments:

The main problem relates to the experimental strategy. Based on a couple of previous microarray studies, the authors state that "the comparison of distinct arrest phenotypes allows the identification of transcripts expressed at specific stages of germ differentiation during normal spermatogenesis". Although this is true (most probably for a high number of genes), they cannot claim that they "unveiled the transcriptional profiles of male germ cell types", or that they used this approach "to fully characterize the transcriptional profiles of human male germ cells". The genes that are up/downregulated in the different pathological conditions cannot be directly attributed to the corresponding cell types in a normal testis context. For instance, Sertoli cells may most probably express some different genes in SCO syndrome than in the presence of germ cells (the importance of the cross-talk should not be underestimated). Likewise, spermatogenic cells that are arrested at a certain stage will most probably present some aberrant gene expression. Particularly, the expression times for some genes may be altered in arrested testes (e.g. normal spermatogonia may express a subset of genes that are not being detected in arrested spermatogonia and therefore will be attributed to spermatocytes, etc.). Moreover, as different pathological conditions with spermatogenic arrest are used for each pairwise comparison, even for the same cell types that are present in both compared conditions, some of the genes that are aberrantly expressed may differ between them. The authors must be aware of the limitations of the approach. In this regard, they must rethink their objective, and the information and conclusions they draw from this study.

Response: We agree with the reviewer and changed our conclusions to less bold statements (See lines 84-89 and 346-352). Indeed the approach we used for this study has some limitations, which we have now discussed in an additional paragraph (lines 335-345). To address the reviewer's important question regarding changes that might be related to the respective germ cell arrest, we additionally analyzed genes putatively associated with the pathological conditions in our samples, which is also in line with a comment by reviewer #1. For the revised version of the manuscript we identified a list of potential infertility misregulated genes (Table S2, and genes shaded in blue in Tables S4-S6), namely the genes upregulated in SCO, SPG, SPC, and SPD samples in comparison to SPG, SPC, SPD, and CTR, respectively and analyzed in silico the molecular pathways enriched in these genes using Reactome analysis. Moreover we analyzed the cell-type specific expression of the 50 most severe genes (according to \log_2FC) of each infertility phenotype in our scRNA-seq dataset (new supplementary figure S6). The results were added to a new results section "Identification of putatively misregulated genes in male infertility" in lines 241-263 and discussed in lines 275-281.

Reviewer #3 comment (2): Minor comments:

-Table 1, and Fig 1B: The term SCO (Sertoli cell-only) is used instead of Sertoli cells. The percentages of spermatogonia and Sertoli cells for the normal tubules are missing.

Response: We would like to clarify that the percentages/cellularities given in Table 1 and Fig. 1B refer to the most advanced state of the tubules, therefore SCO does not correspond to the number of Sertoli cells per tubule but to the tubules containing only Sertoli cells. Moreover, in SCO-classified samples, there were no tubules containing germ cells. In contrast, our histological examination of samples with complete spermatogenesis showed that in 12% of the tubules the most advanced cell states are round spermatids or spermatocytes. Thus there are no spermatogonial arrested and

Sertoli cell-only tubules in the samples with full spermatogenesis. To clarify this point, we revised the description of table 1 and figure 1.

Reviewer #3 (Significance (Required)):

There is wide interest in human testicular transcriptome. Although some reports already exist in the literature, the whole transcriptome of the different spermatogenic cells and its regulation, in particular regarding non-coding RNAs and splice variants, as well as the involvement of the different genes and variants in testicular pathology, is far from being elucidated. My objections regarding the approach and the conclusions that are drawn, were already indicated above.

The expected audience are researchers in the field of gametogenesis and male reproduction. My field of expertise is molecular biology of spermatogenesis.

Response: We highly appreciate the point this reviewer is raising about the significance on assessing the human testicular transcriptome in the different germ cell types for the field and felt free to use similar wording in the revised manuscript (lines 73-77 and 346-352). With our approach, we aimed at tightening the above-mentioned knowledge gap of the field by applying state-of-the-art bioinformatic tools. We thank the reviewer for the highly constructive feedback and incorporated it by adapting our conclusion statements and pointing out the limitations of our study (as mentioned in the response to reviewer #3's comment (1)). We trust that our study will bring more awareness to the need to study transcriptome variation not only in spermatogenesis but also in other (somatic) tissues.

Literature included in the point-by-point-response

Camacho Londoño J, Philipp SE. A reliable method for quantification of splice variants using RT-qPCR. *BMC Molecular Biol* 2016;**17**:8.

Dick F, Nido GS, Alves GW, Tysnes O-B, Nilsen GH, Dölle C, Tzoulis C. Differential transcript usage in the Parkinson's disease brain. In Hamilton BA, editor. *PLoS Genet* 2020;**16**:e1009182.

Hermann BP, Cheng K, Singh A, Roa-De La Cruz L, Mutoji KN, Chen I-C, Gildersleeve H, Lehle JD, Mayo M, Westernströer B, *et al.* The mammalian spermatogenesis single-cell transcriptome, from spermatogonial stem cells to spermatids. *Cell Reports* 2018;**25**:1650-1667.e8.

Houston BJ, Riera-Escamilla A, Wyrwoll MJ, Salas-Huetos A, Xavier MJ, Nagirnaja L, Friedrich C, Conrad DF, Aston KI, Krausz C, *et al.* A systematic review of the validated monogenic causes of human male infertility: 2020 update and a discussion of emerging gene–disease relationships. *Human Reproduction Update* 2021;**0**:15.

Mahyari E, Guo J, Lima AC, Lewinsohn DP, Stendahl AM, Vigh-Conrad KA, Nie X, Nagirnaja L, Rockweiler NB, Carrell DT, *et al.* Comparative single-cell analysis of biopsies clarifies pathogenic mechanisms in Klinefelter syndrome. *The American Journal of Human Genetics* 2021;**108**:1924–1945.

Marques-Coelho D, Iohan L da CC, Melo de Farias AR, Flaig A, Lambert J-C, Costa MR. Differential transcript usage unravels gene expression alterations in Alzheimer's disease human brains. *npj Aging Mech Dis* 2021;**7**:2.

Solovyeva EM, Ibebunjo C, Utzinger S, Eash JK, Dunbar A, Naumann U, Zhang Y, Serluca FC, Demirci S, Oberhauser B, et al. New insights into molecular changes in skeletal muscle aging and disease: Differential alternative splicing and senescence. *Mechanisms of Ageing and Development* 2021;**197**:111510.

Suntsova M, Gaifullin N, Allina D, Reshetun A, Li X, Mendeleeva L, Surin V, Sergeeva A, Spirin P, Prassolov V, et al. Atlas of RNA sequencing profiles for normal human tissues. *Sci Data* 2019;**6**:36.

Tekath T, Dugas M. Differential transcript usage analysis of bulk and single-cell RNA-seq data with DTUrtle. *Bioinformatics* 2021;btab629.

Wyrwoll MJ, Köckerling N, Vockel M, Dicke A-K, Rotte N, Pohl E, Emich J, Wöste M, Ruckert C, Wabschke R, et al. Genetic architecture of azoospermia—Time to advance the standard of care. *European Urology* 2022;

October 7, 2022

Re: Life Science Alliance manuscript #LSA-2022-01633-TR

Dr. Sandra Laurentino
University of Münster
Centre of Reproductive Medicine and Andrology, Institute of Reproductive and Regenerative Biology
Domagkstr. 11
Münster, NRW 48149
Germany

Dear Dr. Laurentino,

Thank you for submitting your revised manuscript entitled "Exploring gene expression and isoform usage during human spermatogenesis" to Life Science Alliance. The manuscript has been seen by the original reviewers whose comments are appended below. While the reviewers continue to be overall positive about the work in terms of its suitability for Life Science Alliance, some important issues remain.

Our general policy is that papers are considered through only one revision cycle; however, given that the suggested changes are relatively minor, we are open to one additional short round of revision. Please note that I will expect to make a final decision without additional reviewer input upon re-submission. Please consider Reviewer 1's final comments below. Since the data requests were not made previously, we will not require that these are performed. However, if you have the data and would like to add it, this would be a welcome addition. The textual recommendations should be addressed.

Please submit the final revision within one month, along with a letter that includes a point by point response to the remaining reviewer comments.

To upload the revised version of your manuscript, please log in to your account: <https://lsa.msubmit.net/cgi-bin/main.plex>
You will be guided to complete the submission of your revised manuscript and to fill in all necessary information.

B. MANUSCRIPT ORGANIZATION AND FORMATTING:

Sincerely,

Reviewer #1 (Comments to the Authors (Required)):

Siebert-Kuss et al. have done a good job of revising their MS in response to reviewer concerns. Thus, I have high enthusiasm for this manuscript, which I think will be a landmark paper on the topic of the intersection between the infertility clinic and basic mechanisms of spermatogenesis. That said, there are some remaining issues:

A. A highlight of this MS is the novel protein markers the authors identified (Fig. 3L). The specificity of these markers needs to be investigated by co-staining with well-established markers. This is important, not only because it is essential to define their specificity but because some of the results are potentially problematic. For example, the claimed spermatogonial marker, LUZP4, appears to label some Sertoli cell nuclei. The claimed spermatogonial marker, TSPY4, appears to be very broad in its specificity; its specificity for undifferentiated vs. differentiating spermatogonia should be investigated.

B. The authors identified 4 interesting genes that are upregulated in SCO samples. They should verify this regulation by in situ hybridization (e.g., RNAscope) or qPCR; merely searching a database is not sufficient.

C. The current title is underwhelming and does not adequately convey the main points of this MS. For example, the word "exploring" sounds weak. It is important to convey that SCO patient transcriptomes were analyzed. How about something like: "Transcriptome analysis of Sertoli-cell only patients reveal..."

D. Abstract. Since this is the one thing that most people will read, it needs to be in high form. At present, it is not. Some suggested improvements. First, sentence 1 is unnecessary and the second half of the sentence 2 is arguably incorrect. Why not instead start with a sentence about male infertility, since that is the nature of the samples examined? Alternatively, some other general but directly relevant topic sentence? The phrase "total RNA-seq" in sentence 3 is incorrect as used. Change sentence to something like: "We compared the transcriptomes of testes from infertile patient with...." Sentence 4 has too much detail; even if one wanted to provide the precise number of DEGs and DTUs, what the range refers to is not spelled out. It is suggested to instead say that thousands of DEGs were identified between the infertile stages, and discuss the DTUs separately (in a conceptual way) in another sentence. Sentence 5 (line 36) makes a minimally important point in this reviewer's opinion, so consider omitting. Consider splitting the last sentence into 2 or more sentences, and say more about the markers, including that the proteins encoded by these genes were shown to have specificity for such and such stage(s).

E. The "Genetic Characterization of the patient cohort" section. The first paragraph of this section does not relate to the RNA-seq data, so it should be made a separate section that comes first in the Results section. The contents of the second paragraph should be moved to the section entitled "Testicular phenotypes are recapitulated..." and the alternative splicing section, as appropriate.

F. Writing. There are writing issues throughout the manuscript, only some of which I will mention. The sentence starting on 84 is awkward and too long. The word "putatively" in the section title on line 241, while strictly speaking correct, sounds awkward in this context. How about naming this section: "Identification of putative infertility genes"? Line 244: add "to" between "compared" and "SPG". Line 247: change to something like: "...processes; e.g. MECP regulates the transcription of genes...". Line 255: change to something like: "...the DEGs with largest regulation...". Line 328: intron retention is brought up here without any mention of how SYCP3 regulation relates to intron retention.

Reviewer #2 (Comments to the Authors (Required)):

The authors have answered all my issues adequately by adding new data, new analyses, and additional text.

Dear Dr. Sawey,

Thank you once again for your continued interest in our manuscript and also to the reviewers for their critical appraisal and enthusiasm for this work. Although you have suggested that we perform only textual changes, we highly appreciated the remaining remarks from reviewer #1 and managed to address all except one within the short time available for this revision. You will find our point-by-point response below (the original comments by the reviewers are in shaded italic text while our responses are in plain text). In response to the suggestion from reviewer #1 we updated the title of the manuscript, which is now “Transcriptome analyses in infertile men reveal germ cell-specific expression and splicing patterns” and performed general language editing with the assistance of a professional in order to improve the readability of some sections. We uploaded a new, revised manuscript with the changes highlighted in yellow.

We hope the manuscript is now suitable for publication in LSA.

Point-by-point response to the reviewers' comments

Reviewer #1 (Comments to the Authors (Required)):

Siebert-Kuss et al. have done a good job of revising their MS in response to reviewer concerns. Thus, I have high enthusiasm for this manuscript, which I think will be a landmark paper on the topic of the intersection between the infertility clinic and basic mechanisms of spermatogenesis.

Response: We thank the reviewer for this enthusiastic response to our manuscript. We have addressed the remaining issues below.

That said, there are some remaining issues: A. A highlight of this MS is the novel protein markers the authors identified (Fig. 3L). The specificity of these markers needs to be investigated by co-staining with well-established markers. This is important, not only because it is essential to define their specificity but because some of the results are potentially problematic. For example, the claimed spermatogonial marker, LUZP4, appears to label some Sertoli cell nuclei.

Response: We carefully reevaluated the protein expression profile of LUZP4 in three patients with full spermatogenesis and could not observe any staining in the Sertoli cell nuclei. To better demonstrate the specificity of this marker, we updated the manuscript with new, higher resolution images in Fig 3L and below we provide representative pictures of Sertoli cell nuclei in comparison

with LUZP4-positive spermatogonia in three control patients with full spermatogenesis, as well as the IgG control, demonstrating the specificity of this marker (Response letter Fig. 1).

Response letter Fig. 1: Panels A, B, and C are representative images of three independent patient samples with full spermatogenesis that were immunohistochemically stained for LUZP4, which gives positive signal in a subpopulation of spermatogonia (white arrow heads), whereas Sertoli cell nuclei are negative (red stars). Panel D is the IgG control that shows no staining. Scale bars = 20 μ m.

The claimed spermatogonial marker, TSPY4, appears to be very broad in its specificity; its specificity for undifferentiated vs. differentiating spermatogonia should be investigated.

Response: We agree with the reviewer that it would be interesting to do follow-up analyzes on our newly-identified protein markers, including the specificity for undifferentiated vs. differentiating spermatogonia. Therefore, we co-stained TSPY4 along with the pan-spermatogonial marker MAGEA4 and found $88 \pm 5.2\%$ of the cells were double positive (Response letter Fig. 2A). We then co-stained TSPY4 with the pan-undifferentiated spermatogonial marker UTF1. We found expression of TSPY4 in $85 \pm 5.6\%$ of UTF1-positive cells (Response letter Fig. 2B), indicating TSPY4 is expressed by the majority of spermatogonia and is not specific for undifferentiated spermatogonia. Further investigations, such as co-stainings with additional markers, will be needed in the future to identify the role of TSPY4 in the testis. We included our analyses in the revised version of the manuscript as a new supplementary figure S3. (Lines 180-186, lines 444-463, lines 807-815)

Response letter Fig. 2 (new supplementary Fig. S3): Co-localization analyses of TSPY4 with MAGEA4 and UTF1 in testes with full spermatogenesis. Representative images of testicular tissues stained for TSPY4 (green) together with the pan-spermatogonial marker MAGEA4 (magenta) **(A)** and the pan-undifferentiated spermatogonial marker UTF1 (magenta) **(B)**. The tissue sections were counterstained with DAPI (blue). The IgG control shows no specific immunological staining. Scale bars = 100 μ m (main), 10 μ m (inlays), 50 μ m (IgGs). Boxplots represent the percentage of TSPY4+ cells per 200 MAGEA4+ and UTF1+ cells, respectively, which were quantified in three independent patient samples with full spermatogenesis.

B. The authors identified 4 interesting genes that are upregulated in SCO samples. They should verify this regulation by in situ hybridization (e.g., RNAscope) or qPCR; merely searching a database is not sufficient.

Response: Indeed these are interesting targets for further evaluation, an avenue that we intend to pursue in the future. Due to the limited amount of time given for the revision and the amount of work that we consider would be necessary to properly validate these candidate genes in a larger cohort, we did not perform the suggested analyses.

C. The current title is underwhelming and does not adequately convey the main points of this MS. For example, the word "exploring" sounds weak. It is important to convey that SCO patient

transcriptomes were analyzed. How about something like: "Transcriptome analysis of Sertoli-cell only patients reveal..."

Response: We thank the reviewer for the suggestion to choose a more meaningful title, which we changed to "Transcriptome analyses in infertile men reveal germ cell-specific expression and splicing patterns".

D. Abstract. Since this is the one thing that most people will read, it needs to be in high form. At present, it is not. Some suggested improvements. First, sentence 1 is unnecessary and the second half of the sentence 2 is arguably incorrect. Why not instead start with a sentence about male infertility, since that is the nature of the samples examined? Alternatively, some other general but directly relevant topic sentence? The phrase "total RNA-seq" in sentence 3 is incorrect as used. Change sentence to something like: "We compared the transcriptomes of testes from infertile patient with...." Sentence 4 has too much detail; even if one wanted to provide the precise number of DEGs and DTUs, what the range refers to is not spelled out. It is suggested to instead say that thousands of DEGs were identified between the infertile stages, and discuss the DTUs separately (in a conceptual way) in another sentence. Sentence 5 (line 36) makes a minimally important point in this reviewer's opinion, so consider omitting. Consider splitting the last sentence into 2 or more sentences, and say more about the markers, including that the proteins encoded by these genes were shown to have specificity for such and such stage(s).

Response: We have updated the abstract accordingly. Moreover, we had the help of a language editor to streamline the abstract and the rest of the text as suggested in comment F below. (Lines 29-42)

E. The "Genetic Characterization of the patient cohort" section. The first paragraph of this section does not relate to the RNA-seq data, so it should be made a separate section that comes first in the Results section. The contents of the second paragraph should be moved to the section entitled "Testicular phenotypes are recapitulated..." and the alternative splicing section, as appropriate.

Response: We reorganized the mentioned results sections as suggested. (Lines 93-132)

F. Writing. There are writing issues throughout the manuscript, only some of which I will mention. The sentence starting on 84 is awkward and too long. The word "putatively" in the section title on line 241, while strictly speaking correct, sounds awkward in this context. How about naming this section:

"Identification of putative infertility genes"? Line 244: add "to" between "compared" and "SPG". Line 247: change to something like: "...processes; e.g. MECP regulates the transcription of genes...". Line 255: change to something like: "...the DEGs with largest regulation...". Line 328: intron retention is brought up here without any mention of how SYCP3 regulation relates to intron retention.

Response: Based on this comment, we hired a professional, native-speaking scientific language editor, who screened and optimized the whole manuscript. We highlighted the edited sections in yellow.

Reviewer #2 (Comments to the Authors (Required)):

The authors have answered all my issues adequately by adding new data, new analyses, and additional text.

Response: We appreciate the reviewer's appraisal of the successful revision process.

November 3, 2022

RE: Life Science Alliance Manuscript #LSA-2022-01633-TRR

Dr. Sandra Laurentino
University of Münster
Centre of Reproductive Medicine and Andrology, Institute of Reproductive and Regenerative Biology
Domagkstr. 11
Münster, NRW 48149
Germany

Dear Dr. Laurentino,

Thank you for submitting your revised manuscript entitled "Transcriptome analyses in infertile men reveal germ cell-specific expression and splicing patterns". We would be happy to publish your paper in Life Science Alliance pending final revisions necessary to meet our formatting guidelines.

- please add ORCID ID for secondary corresponding author-they should have received instructions on how to do so
- please add the Abstract in our system

A. FINAL FILES:

B. MANUSCRIPT ORGANIZATION AND FORMATTING:

****It is Life Science Alliance policy that if requested, original data images must be made available to the editors. Failure to provide**

original images upon request will result in unavoidable delays in publication. Please ensure that you have access to all original data images prior to final submission.**

The license to publish form must be signed before your manuscript can be sent to production. A link to the electronic license to publish form will be sent to the corresponding author only. Please take a moment to check your funder requirements.

Sincerely,

November 8, 2022

RE: Life Science Alliance Manuscript #LSA-2022-01633-TRRR

Dr. Sandra Laurentino
University of Münster
Centre of Reproductive Medicine and Andrology, Institute of Reproductive and Regenerative Biology
Domagkstr. 11
Münster, NRW 48149
Germany

Dear Dr. Laurentino,

Thank you for submitting your Resource entitled "Transcriptome analyses in infertile men reveal germ cell-specific expression and splicing patterns". It is a pleasure to let you know that your manuscript is now accepted for publication in Life Science Alliance. Congratulations on this interesting work.

DISTRIBUTION OF MATERIALS:

Again, congratulations on a very nice paper. I hope you found the review process to be constructive and are pleased with how the manuscript was handled editorially. We look forward to future exciting submissions from your lab.

Sincerely,
